ecology

variability, stock–recruitment relationship, density-dependence, density-independence, environmental noise, fishing

**Authors for correspondence:**
Leonie Färber
e-mail: leonie.faerber@uni-hamburg.de
Joël M. Durant
e-mail: joel.durant@ibv.uio.no

†Present address: Institute for Marine Ecosystem and Fisheries Science, Center for Earth System Research and Sustainability (CEN), University of Hamburg, Große Elbstraße 133, 22767 Hamburg, Germany.
‡Present address: Department of Biology and Ecology of Fishes, Leibniz-Institute of Freshwater Ecology and Inland Fisheries, Müggelseedamm 310, 12587 Berlin, Germany.

# Population variability under stressors is dependent on body mass growth and asymptotic body size

Leonie Färber[1,†], Rob van Gemert[2,‡],
Øystein Langangen[1], Joël M. Durant[1]
and Ken H. Andersen[2]

[1]Centre for Ecological and Evolutionary Synthesis (CEES), Department of Biosciences, University of Oslo, PO Box 1066 Blindern, NO-0316 Oslo, Norway
[2]Centre for Ocean Life, National Institute of Aquatic Resources (DTU-Aqua), Technical University of Denmark, Kemitorvet, Building 202, 2800 Kgs Lyngby, Denmark

 LF, 0000-0002-8587-3438; ØL, 0000-0002-6977-6128; JMD, 0000-0002-1129-525X; KHA, 0000-0002-8478-3430

The recruitment and biomass of a fish stock are influenced by their environmental conditions and anthropogenic pressures such as fishing. The variability in the environment often translates into fluctuations in recruitment, which then propagate throughout the stock biomass. In order to manage fish stocks sustainably, it is necessary to understand their dynamics. Here, we systematically explore the dynamics and sensitivity of fish stock recruitment and biomass to environmental noise. Using an age-structured and trait-based model, we explore random noise (white noise) and autocorrelated noise (red noise) in combination with low to high levels of harvesting. We determine the vital rates of stocks covering a wide range of possible body mass (size) growth rates and asymptotic size parameter combinations. Our study indicates that the variability of stock recruitment and biomass are probably correlated with the stock's asymptotic size and growth rate. We find that fast-growing and large-sized fish stocks are likely to be less vulnerable to disturbances than slow-growing and small-sized fish stocks. We show how the natural variability in fish stocks is amplified by fishing, not just for one stock but for a broad range of fish life histories.

## 1. Introduction

The natural environmental conditions, e.g. sea temperature, that a fish stock experiences fluctuate continuously. During their early-life stages, fish are particularly sensitive to environmental

fluctuations, and this sensitivity can result in large variability in the size of the spawning stock and the stock available for fisheries. In addition to these fluctuations in the environment, fish stocks are also subject to fishing pressure, which can, for instance, alter the population structure in terms of age or size [1].

Thus, the combined effects of climate variability and fishing can strongly alter the population dynamics. For instance, when fishing targets large and old individuals, the resilience of the stock to buffer environmental effects might be disturbed [1–4]. Hidalgo *et al.* [4] showed in a simulation of population dynamics that, with exploitation, the importance of the contribution of recruitment to the total biomass increases; accordingly, environmental effects (simulated by white and red noise) have a stronger effect on biomass dynamics. Owing to their longevity, large-sized and slow-growing fish often have many cohorts contributing to the overall recruitment. Thus, poor reproductive output of one weak cohort can be substituted by the other cohorts, thereby buffering environmental fluctuations [3–6]. However, it is also possible that the variability in recruitment will be echoed by a generational lag throughout the stock [7,8]. This phenomenon, which is called 'cohort resonance', has been demonstrated in three species with differing longevity: coho salmon (*Oncorhynchus kisutch*), Pacific hake (*Merluccius productus*) and Pacific Ocean perch (*Sebastes alutus*) [7,8]. Additionally, the effects of environmental conditions experienced in early life are evident throughout the lifespan of the cohort ('cohort effect' [9–11], e.g. in capelin (*Mallotus villosus*) [10]. Studies have shown that traits that promote slow population increase and low turnover rate, such as late maturation, slow body mass (size) growth rate and large asymptotic size, leave a stock more vulnerable to exploitation [5,7,12,13]. A study on North Atlantic cod (*Gadus morhua*) stocks showed that environmental effects (autocorrelated and non-correlated dynamics) that influence recruitment by modulating, for example, food availability and temperature lead to increased sensitivity to collapse among fished populations [14]. Further, species with high rates of body mass growth are more prone to collapse under fishing pressure and environmental variability [15].

In this study, we systematically analyse a broad spectrum of life histories of fish species in response to environmental impacts and fishing pressure in order to draw general conclusions about fish life histories and their sensitivity towards disturbances. For this large-scale exploration of fish dynamics, we use a trait-based description of fish diversity, focusing on two traits: asymptotic body mass (size) and body mass growth rate [16]. Focusing on asymptotic size enables the generalization of the variety of fish life histories by characterizing different species by a single trait [17]. The body mass growth rate further distinguishes between slow- and fast-growing species of similar asymptotic size. Using this approach, we are able to investigate the dynamics of fish stocks ranging from slow to fast growth, and from small to large asymptotic size.

Understanding the response of fish populations to disturbances helps to direct sustainable management efforts [3,4]. In this study, we test the sensitivity of fish populations to random white noise and to autocorrelated red noise. True environmental conditions involve red noise more frequently than white, especially in the marine environment [18,19]. However, the North Atlantic Oscillation dynamics from 1822 to 2000 can be approximated by white noise [20]. Red noise potentially leads to large variability in the stock's dynamics [21] and extinction owing to the higher probability of prolonged periods of bad conditions [22]. Here, we examine the impact of environmental white and red noise on recruitment efficiency and explore the effect of recruitment variability on spawning stock biomass, across a broad range of fish life histories, in order to identify general patterns. In particular, we investigate how early-life density dependence, in the form of a stock–recruitment relationship, buffers these varying environmental effects in fished populations. Additionally, we systematically explore the roles played by growth and body size in population dynamical responses to noise colour and mortality for a broad range of fish life histories, and demonstrate that the natural variability in fish stocks is amplified by fishing.

## 2. Material and methods

We used an existing size-based model for demographic characteristics and recruitment efficiency, as described previously by Andersen & Beyer [23]. Even though the model is size based, we formulated the present model in terms of age because the effect of noise on recruitment is an inherently annual process (see tables 1 and 2 for equations and parameters, respectively; electronic supplementary material, figure S1 for sensitivity analysis of the choices of parameter values). The age-structured model was solved as a classic matrix model (e.g. [24]). The environmental variations (noise) affect

**Table 1.** Equations used for the calculation of the model. Index $i$ refers to age and $t$ to time.

| function | equation | no. |
|---|---|---|
| von Bertalanffy weight at age $i$ | $w_i = W_\infty(1 - e^{-(1-n)AW_\infty^{n-1}i})^{1/(1-n)}$ | (2.1) |
| abundance | $N_{i,t+1} = S_{i-1}N_{i-1,t}$ <br> $N_{1,t+1} = R_t$ | (2.2) |
| survival | $S_i = e^{-(\mu_{p,i} + \mu_{F,i})}$ | (2.3) |
| natural mortality | $\mu_{p,i} = aAw_i^{n-1}$ | (2.4) |
| fishing with trawl selectivity | $\mu_{F,i} = F\left[1 + \left(\dfrac{w_i}{\eta_F W_\infty}\right)^{-u_F}\right]^{-1}$ | (2.5) |
| maturity | $\psi_i = \left[1 + \left(\dfrac{w_i}{\eta_m W_\infty}\right)^{-u_m}\right]^{-1}$ | (2.6) |
| recruitment | $R_t = R_{max} \dfrac{\alpha_t P_{w_{egg} \to w_R} B_{SSB,t}}{R_{max} + \alpha_t P_{w_{egg} \to w_R} B_{SSB,t}}$ | (2.7) |
| egg production | $\alpha_t = \dfrac{A e^{x_t} \varepsilon_r (1 - \varepsilon_a) W_\infty^{n-1}}{w_{egg}}$ | (2.8) |
| survival from egg to recruitment | $P_{w_{egg} \to w_R} = \left(\dfrac{w_R}{w_{egg}}\right)^{-a}$ | (2.9) |
| spawning stock biomass | $B_{SSB,t} = \displaystyle\sum_{i=1}^{m-1} \psi_i N_{i,t} w_i$ | (2.10) |
| white noise | $x_{white,t} \sim \text{Norm}(0, \sigma^2)$ | (2.11) |
| red noise | $x_{red,t} = c\, x_{red,t-1} + x_{white,t}(1 - c^2)^{1/2}$ <br> $x_{red,1} = x_{white,1}$ | (2.12) |

recruitment through a density-independent mortality that acts on age-0 fish (figure 1). In addition, we induced pressure on the populations by adding fishing pressure (figure 1).

We investigated a parameter space within the 80% most common asymptotic sizes and body mass growth rates of the species included in the *FishBase* database [25], available through the R [26] (v. 3.4.4) package '*rfishbase*' (v. 3.0.4) [27]. We downloaded the available data in July 2019. We used a combination of 1600 log-transformed weight and growth rates, where asymptotic length $L_\infty$ was converted to asymptotic weight $W_\infty$ and the growth parameter $K$ to the growth coefficient used in this model, $A$ (electronic supplementary material, Methods), where each combination can be considered to represent a fish population with a specific life history. The coefficient $A$ represents processes that involve gain and assimilation of energy, and is derived from the von Bertalanffy growth equation [28]. $A$ can be related to the 'classical' von Bertalanffy growth parameters $K$ and asymptotic length $L_\infty$ with a metabolic scaling factor and weight at age $w_i$ (equation (2.1), table 1). We use $n = \frac{3}{4}$ [29] as the metabolic scaling exponent because it takes into account the digestive and respiratory surface area within fish scales with a higher exponent than 2/3 [16,23]. Asymptotic weight ranged from 1 g to 48 kg, and growth rate $A$ ranged from 1.2 $\text{g}^{1-n}$ year$^{-1}$ to around 41 $\text{g}^{1-n}$ year$^{-1}$ (cf. [16]; see also electronic supplementary material, figure S2).

For each population, we calculated the abundances through time (equation (2.2), table 1, repeated below). Each year, $N_{i,t+1}$ surviving individuals of weight $w_i$ enter the next age class and reproduces when mature.

$$N_{1,t+1} = R_t \text{ and } N_{i,t+1} = S_{i-1}N_{i-1,t}, \tag{2.2}$$

**Table 2.** Parameters used in the equations, noted in table 1. Index $i$ refers to age and $t$ to time. Justification of parameter values and analysis of sensitivity of the results to changes in values can be found in the electronic supplementary material, appendix.

| symbol | description | value | unit |
|---|---|---|---|
| $A$ | body mass growth rate | stock-specific | $g^{1-n}\,year^{-1}$ |
| $W_\infty$ | asymptotic size (body mass) | stock-specific | g |
| $n$ | metabolic exponent | 3/4 | — |
| $a$ | physiological mortality | 0.35 | — |
| $F$ | maximum fishing mortality | variable | $year^{-1}$ |
| $\eta_F$ | start of fishing relative to $W_\infty$[a] | 0.15 | — |
| $u_F$ | width of trawl selectivity switching function[b] | 3 | — |
| $\eta_m$ | size at maturation relative to $W_\infty$ | 0.25 | — |
| $u_m$ | width of maturity switching function | 10 | — |
| $R_{max}$ | maximum recruitment | 1 | Numbers/time |
| $\varepsilon_r$ | recruitment efficiency | 0.1 | — |
| $w_R$ | weight at recruitment age | stock-specific weight at age 1 | g |
| $w_{egg}$ | egg weight | 0.001 | g |
| $\varepsilon_a$ | fraction of energy used for activity | 0.8 | — |
| $m$ | number of age classes | 100 | — |
| $x$ | noise (white noise or red noise) | variable | — |
| $\sigma$ | white noise standard deviation[c] | 0.7 | — |
| $c$ | red noise correlation coefficient[c] | 0.9[d] | — |

[a]Adjustment from Andersen and Beyer [23] with an earlier start of fishing relative to $W_\infty$.
[b]Adjustment from Andersen and Beyer [23] with a less sharp trawl selectivity.
[c]Novel parameters not included in the model parameters from Andersen and Beyer [23].
[d]Corresponding to approximately a cycle of 9 years (autocorrelation value < 0.2 at lag of 9 years).

where $N_{i,t}$ is the abundance at age $i$ and year $t$, and the weight-specific survival $S_i$ (equation (2.3), table 1) is calculated from natural mortality (equation (2.4), table 1) and fishing mortality. Fishing mortality affects individuals according to a sigmoidal weight-specific trawl selectivity (equation (2.5), table 1). The entry size to the fishery was, slightly before maturity, at $\eta_F W_\infty$ (with $\eta_F = 0.15$, equation (2.5), tables 1 and 2), whereas size at maturity was $\eta_m W_\infty$ (with $\eta_m = 0.25$, equation (2.6), tables 1 and 2). The first age class ($N_{1,t}$) is determined by the recruitment $R_t$. Each population was divided into a total of $m = 100$ age classes. This number of age classes is sufficient for even large-sized and slow-growing species to reach a size close to the asymptotic size. Thus, by setting a high number of age classes, we avoid an involuntary eviction of the old fish from the model. Shorter lived species will have abundances very close to 0, in these older age classes, owing to high mortality in early years; therefore, the results will not be biased.

Recruitment $R_t$ (equation (2.7), table 1) was calculated using a Beverton–Holt stock–recruitment relationship [30]. Recruitment is regulated by density-independent factors (environmental noise) and density-dependent factors (spawning stock biomass, $B_{SSB,t}$) (equation (2.7), table 1 and figure 1). As described by equation (2.7) in table 1, $R_{max}$ is the maximum recruitment (here, arbitrarily set to 1 per year); $\alpha_t$ the egg production per unit biomass at time $t$ (equation (2.8), table 1); $P_{w_{egg} \to w_R}$ the survival from egg ($w_{egg}$) to recruitment size $w_R$ here at age 1 ($w_R = w_1$) (equation (2.9), table 1) and $B_{SSB,t}$ is the spawning stock biomass at time $t$ (equation (2.10), table 1). The egg production per unit biomass $\alpha_t$ is a function of recruitment efficiency $\varepsilon_R$ (the cost of reproduction and survival from spawned eggs [23]), asymptotic weight, $W_\infty$, growth rate, $A$ and the fraction of energy assigned for activity and not reproduction, $\varepsilon_a$ (equation (2.8), table 1). Because it is difficult to obtain recruitment efficiency, $\varepsilon_R$, values from the literature, we used an estimation described by Hartvig *et al.* [31] (equation (2.8), table 1), which is based on calculations from a bioenergetics model and empirical measurements (see also electronic supplementary material). In this case, $\varepsilon_R$ was affected by a lognormally distributed environmental noise $x_t$, leading to a time-varying recruitment efficiency $\varepsilon_R e^{x_t}$.

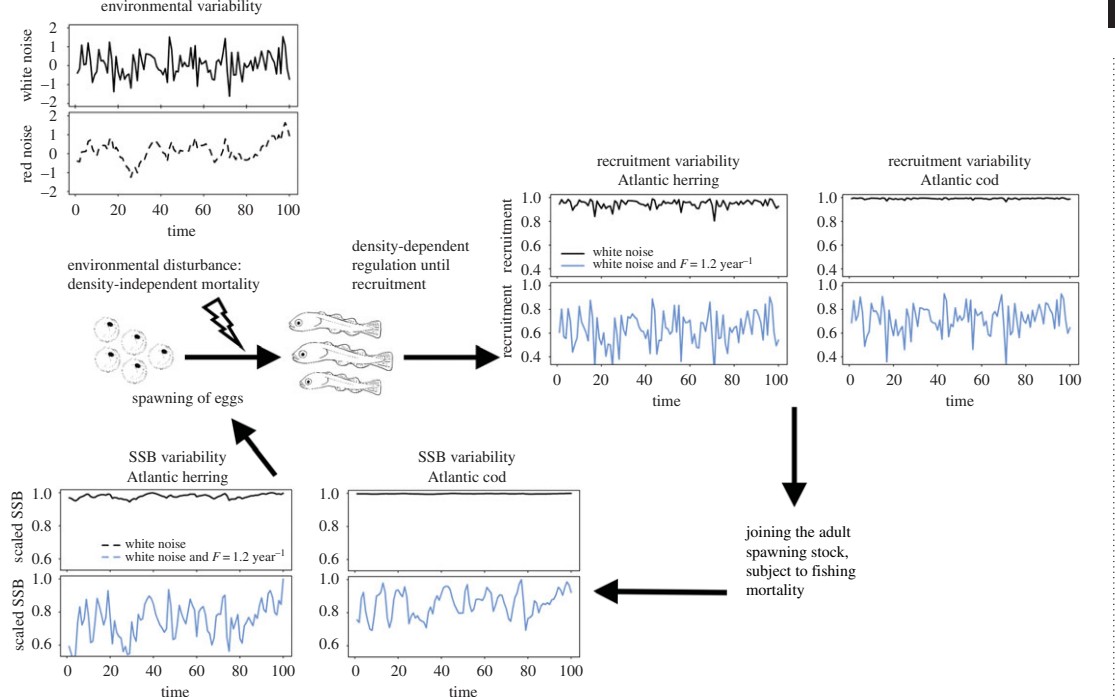

**Figure 1.** Overview of the processes in the model. The spawning stock biomass ($B_{SSB}$; here, scaled to the maximum $B_{SSB}$) produces a certain amount of eggs. Then, environmental noise (upper left plots: random white noise (solid line); autocorrelated red noise (dashed line)) affects the survival of eggs, which leads to density-independent mortality. The remaining eggs must survive until recruitment (here, at age 1), which is regulated by density-dependent processes. The effects of environmental variability were detectable on recruitment efficiency (right plot; here, only plotted for white noise, because there was no significant difference between red noise and white noise scenarios ($p$-value $> 0.05$, figure 3)). Smaller species (such as Atlantic herring, left-side plots) have a lower buffer capacity for environmental disturbances, and thus show higher variability than large-sized species such as Atlantic cod (right-side plots). If fishing is carried out (blue line; here, with $F = 1.2$ year$^{-1}$, black line $F = 0$ year$^{-1}$) the fish enter into fishery at a size $\eta_F W_\infty$ (see tables 1 and 2), which reduces the stock's biomass.

We ran our population model in a white (equation (2.11), table 1) or red noise environment (equation (2.12), table 1) at three different levels of fishing mortality $F = 0$, $F = 0.3$ and $F = 1.2$ year$^{-1}$) (equation (2.5), table 1).

In order to ensure that the variability in recruitment was comparable between species, we calculated the coefficient of variation (CV) for the recruitment values over 499 years. We excluded populations with unrealistic low recruitment (with a mean recruitment remaining under 0.01) as they did not yield reliable results.

For illustrative purposes, we selected five species that covered the range of asymptotic sizes and body mass growth rates among commercially relevant fish stocks (table 3 and figure 2). We calculated the mean body mass growth rate and asymptotic body size (converted from the length and growth parameters in *FishBase*) and then found the nearest matching value in our model range. Thus, the selected species show average values for several populations, based on information contained in the *FishBase* database and not data for individual fish (see electronic supplementary material, figure S2, for the data on the populations).

The Sebastidae family (rockfish) represents slow-growing, relatively large-sized fish. The four other families contain species that are among the most intensely harvested [32] and, therefore, of high commercial interest.

## 3. Results

The environmental noise affected the stock by causing variable survival at the egg stage (figure 1). This variability in egg survival and consequently recruitment was then translated to the whole stock into an increased variability of the spawning stock biomass (figure 1; electronic supplementary material, figure S3).

Slower growing species and species with smaller asymptotic size were found to be more sensitive in terms of the effects of noise on recruitment (figure 2; electronic supplementary material, figure S4). As

**Table 3.** Overview of the growth rates and asymptotic weights used in the model for the five example species. Divergence from the reported rates in *FishBase* due to our selected parameter space and identification of the closest value to the reported mean.

| species | growth rate $A$ in model [$g^{0.25}$ year$^{-1}$] | mean growth rate $A$ from *FishBase* [$g^{0.25}$ year$^{-1}$] | asymptotic weight in model [kg] | mean asymptotic weight from *FishBase* [kg] |
|---|---|---|---|---|
| Atlantic herring *Clupea harengus* (Clupeidae) | 5.2 | 5.4 | 0.4 | 0.4 |
| Atlantic cod *Gadus morhua* (Gadidae) | 5.6 | 5.7 | 12.0 | 12.8 |
| yellowfin tuna *Thunnus albacares* (Scombridae) | 22.0 | 23.0 | 47.8 | 53.8 |
| golden redfish *Sebastes norvegicus* (Sebastidae) | 1.7 | 1.8 | 1.7 | 1.7 |
| Peruvian anchoveta *Engraulis ringens* (Engraulidae) | 14.0 | 14.2 | 0.05 | 0.05 |

such, species that were both small in size (with an asymptotic size $W_\infty < 2\,g$) and slow growing (with growth rates $A < 2.7\,g^{0.25}$ year$^{-1}$) collapsed in all the scenarios (figure 2; electronic supplementary material figure, S4). In general, over all scenarios, slow-growing species ($A < 3\,g^{0.25}$ year$^{-1}$) were significantly more susceptible ($p < 0.001$) to disturbances than fast-growing and large-sized species (e.g. yellowfin tuna $A = 22\,g^{0.25}$ year$^{-1}$) (figures 2 and 3; electronic supplementary material, figure S4). Cod showed lower variability in recruitment than small-sized species such as herring (figure 3; electronic supplementary material, figures S3 and S4) despite having similar growth rates (table 3). This suggests that large-sized species can sustain environmental disturbances well.

Fishing mortality generally increased recruitment variability across all species, although the magnitude of increase differed for different species. For instance, when fished at high pressure, cod recruitment variability increased to a level similar to that of herring (figure 3). For slow-growing species, such as the golden redfish (figures 2 and 3), increasing fishing pressure caused a significant strong increase in recruitment variability ($p < 0.001$), with higher fishing pressure leading to a collapse. At high fishing pressure (i.e. fishing mortality of 1.2 year$^{-1}$), more stocks collapsed and recruitment variability increased, although, in general, fast-growing species still showed little recruitment variability (figure 2; electronic supplementary material, figures S4 and S5).

There were no clear differences between the variability induced by white or red noise (figure 3; electronic supplementary material, figure S4). However, a slightly higher proportion of stocks collapsed under red noise than under white noise in all scenarios (electronic supplementary material, figure S4). The differences between white and red noise scenarios were mainly observed for the slower growing species. This trend was applicable across all asymptotic sizes, and the differences increased with increasing fishing pressure (electronic supplementary material, figure S4).

## 4. Discussion

We illustrate how a model accounting for only two traits, the asymptotic size and growth rate, can be used to systematically quantify the sensitivity of fish stocks to environmental variability and fishing pressure. Frequently, larval survival and, accordingly, recruitment success is directly or indirectly influenced by the environment through factors such as food availability [33], drifting to favourable nursery habitats [34,35] and the effect of temperature on growth [36]. However, species with an asymptotic weight of around 140 g and a growth rate of $> 3\,g^{0.25}$ year$^{-1}$ showed less than 20% variability ($\log_{10} = -0.7$) in their recruitment in the present simulation. This indicates that large-sized species are better able to buffer environmental effects than small-sized and slower growing species (figure 2; electronic supplementary material, figure S4). The larger and/or faster growing the species, the more weakly it was found to respond to the variability in the environment and the lower the

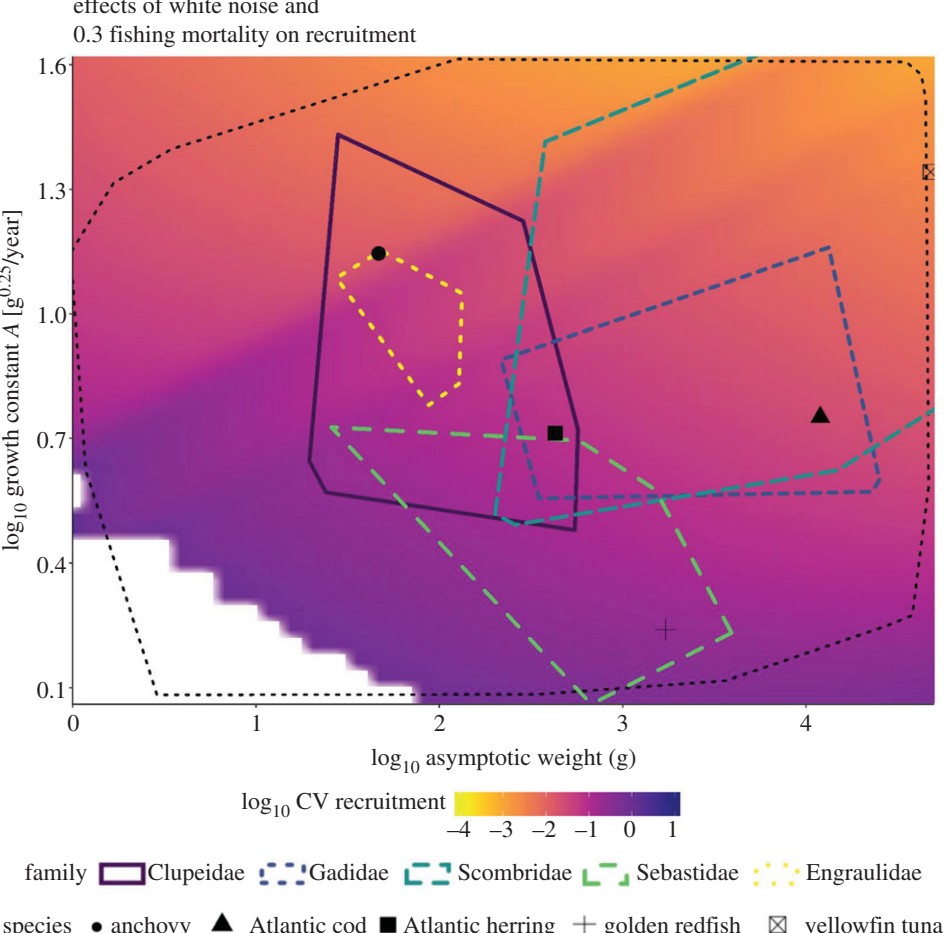

**Figure 2.** Effect of white noise and 0.3 fishing mortality on recruitment efficiency. The logarithm ($\log_{10}$) of the coefficient of variation (CV) as a measure of variability over the whole parameter space for asymptotic size $W_\infty$ and body mass growth rate $A$ are shown for the scenario with white noise and 0.3 year$^{-1}$ fishing mortality. The darker the colour, the higher the variability. The white areas indicate stock collapse. We plotted the parameter space for species from five fish families; data were obtained from the growth database contained in the R-package 'rfishbase' and converted to $W_\infty$ and $A$-values (see electronic supplementary material for details). Five species of these families are indicated according to their position in the parameter space: anchovy, Atlantic cod, Atlantic herring, golden redfish and yellowfin tuna (table 3). The black dotted line indicates a convex hull fitted to the *FishBase* data, indicating the range of biological realistic parameter combinations. Figures for the other scenarios did not differ qualitatively from the one presented here and are presented in the electronic supplementary material, figure S4.

degree of fluctuation in its recruitment (figure 2; electronic supplementary material, figure S4). For the large-sized and fast-growing species the recruitment remains stable (electronic supplementary material, figure S5). This could result from the inherent characteristic of large-sized species to exhibit increased longevity, with multiple cohorts contributing to reproduction. In addition, fast-growing species have high turnover rates, reaching maturity rapidly and contributing to reproduction [37].

The increased contribution of multiple cohorts to reproduction might buffer the effects of environmental variability by spreading spawning over a longer period throughout the season; this bet-hedging strategy mitigates the impacts of unfavourable environmental conditions such as exposure to currents and mismatch with their prey at the larval stage [6,38,39]. Indeed, in the absence of fishing, we show that large-sized species are more resilient to changes in the environment and show less fluctuation in their recruitment in response to year-to-year variability in the environment, as well as in response to longer lasting effects represented by the red noise scenarios (figure 3). However, we find, as predicted by several other studies (e.g. [3,40,41]), that the erosion of the stock structure (e.g. by fishing) may lead to reduced capacity of the stock to buffer disturbances. Consequently, resilience is eroded and variability in recruitment increases. However, fast-growing and large-sized species are more capable of buffering against these effects (electronic supplementary material, figure S5).

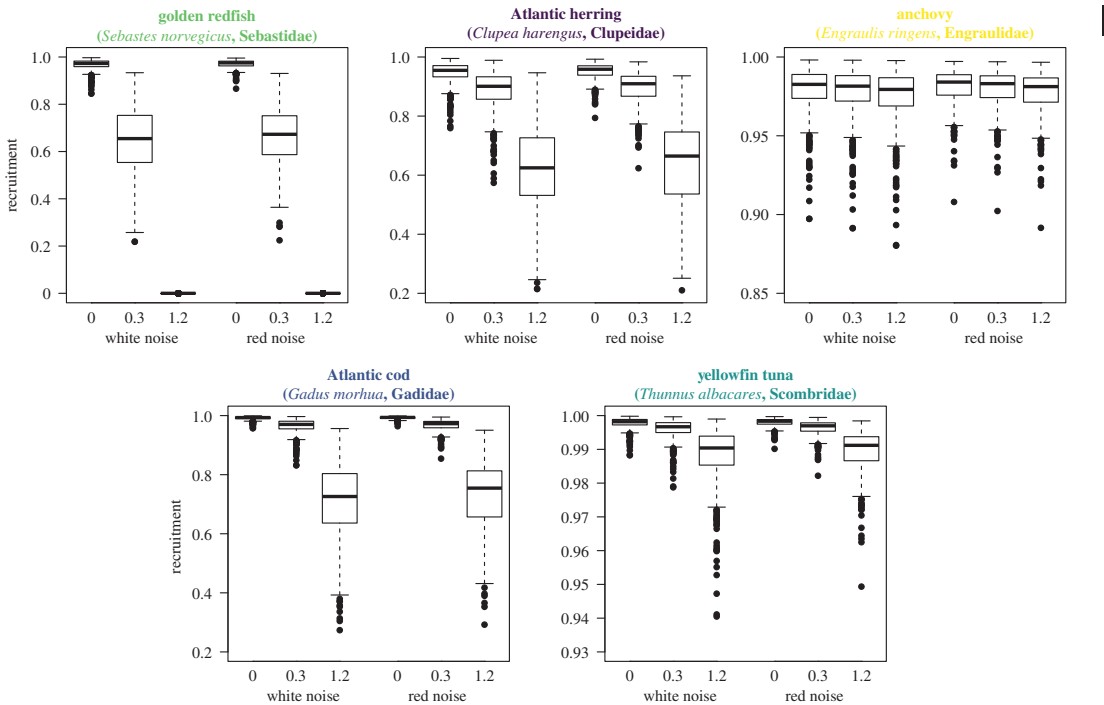

**Figure 3.** Response of recruitment to environmental disturbance and/or fishing of five selected species with varying life histories within the parameter space of growth and asymptotic size (anchovy, Atlantic cod, Atlantic herring, golden redfish and yellowfin tuna, table 3 for details). Their respective spread in recruitment (can be seen as variability) is plotted for each of the scenarios. The three boxplots on the left of each respective species indicate scenarios with white noise with no fishing, with 0.3 year$^{-1}$ fishing mortality and 1.2 year$^{-1}$ fishing mortality; followed by the three scenarios with red noise. Note the varying $y$-axis scales.

Vert-pre *et al.* [42] have shown that some fish stocks do not show the classical direct link between spawning biomass and recruitment (a stock–recruitment relationship). In this context, the environment seems to be an important driver of recruitment dynamics, as observed in several stocks of fish such as herring and cod [43,44]. In our model, we assumed an inherent stock–recruitment relationship. Therefore, as a result of fishing activity that targets large-sized individuals, the stock's age structure is curtailed and only younger cohorts with small-sized individuals contribute to the spawning stock biomass. This may lead to higher variability in recruitment, especially in slow-growing species (electronic supplementary material, figures S4 and S5). It is often assumed that older females can contribute greater fecundity [45] and larger eggs with better quality (e.g. [46]) to the recruitment, and, thus, that older females might contribute more successfully to the reproductive output (cf. [6,38]). However, we did not specifically model age structure effects on recruitment. In many stocks, the maternal effects associated with older fish do not significantly affect recruitment success [47,48] and hence do not affect the overall abundance of the stock [49]. However, at high spawning stock biomass, the variability in recruitment is strongly dampened through compensatory processes [1,50]. Species with high population biomass due to large asymptotic weight and rapid growth, such as the tuna, probably have to be subject to substantial erosion of age or size structure (i.e. exceeding the value applied in our scenarios of fishing mortality) to show an effect on recruitment variability; however, yellowfin tuna has apparently been subjected to overfishing [51]. Nonetheless, it seems the tuna's recruitment is not strongly affected by a decreased stock biomass (cf. [51]). Cod stocks all over the North Atlantic have been depleted substantially because of overfishing and unfavourable climatic conditions; recovery is often inhibited [14,52,53], although signs of recovery have been reported [54]. Similarly, many small-sized pelagic fishes, such as herring, experience high fishing pressure and, in general, exhibit stronger fluctuation in recruitment with environmental conditions [55]. However, they seem to recover relatively faster by regaining or maintaining a population structure that includes old individuals [56,57].

The present model indicates that the growth rate, in combination with asymptotic size, determines where a stock is situated in the 'variability' space. Larger species are more robust than smaller species; however, if their growth rate is very low, they are unable to counterbalance the combined effects of environmental variability and strong fishing pressure; this leads to an increase in

recruitment variability or even collapse [5,7,12,13]. The importance of growth rate as a buffer against environmental variability can also be seen in various species of the Scombridae family. Fast-growing (tropical) species, such as the yellowfin tuna, show fewer declines in response to disturbances than more temperate species like Atlantic bluefin tuna (*Thunnus thynnus*) with slower growth at similar body size [58]. Thus, the location of a fish stock on the slow–fast continuum of life histories yields extensive insights into its vulnerability (cf. [58]).

We included the impact of environmental noise on the recruitment efficiency in order to investigate the effect of recruitment variability on the spawning stock biomass and examine how early-life density dependence, in the form of a Beverton–Holt stock–recruitment relationship, buffers this effect. The impact of environmental noise on maximum recruitment (representing the early-life carrying capacity in a certain environment) [59], or on the survival of the adult stock, could also have been included, but was not the focus in this study. Increased variability, e.g. in fishing pressure, may have altered the outcomes. For example, in declining stocks, fishing mortality may potentially be intensified at low biomasses as a result of delayed management action. This may lead to increased probability of collapse in stocks (cf. [15]). However, the main response of (marine) fish to environmental variability is evident in the survival of the early-life stages, which then translates into recruitment [7]. Therefore, we limited the present investigation to these dynamics.

In this study, we used a simplified model to obtain generalized results (cf. [60]); consequently, the present model has some limitations, e.g. it does not account for individual variations in growth or for environmental impact on growth. Despite these limitations, we argue that our modelling approach is useful to obtain information on general patterns across fish life histories. However, we acknowledge that not all parameters used in the model fit individual species or populations. We note that a sensitivity analysis (electronic supplementary material, appendix figure A5) indicates that the exact parameter values do not, in general, alter our results qualitatively. However, in the few cases where changes occur, these are stronger for the slow-growing and small-sized classes that experience fishing pressure. These populations already respond strongly to environmental variability and fishing pressure.

In our model, the noise colour did not lead to strong differences in the response of the stocks, although slightly more stocks collapsed or displayed larger variability under red noise conditions, which is consistent with the literature (e.g. [14,20–22]). Slower growing species in the scenarios with fishing reacted especially strongly to red noise. This may potentially be attributable to the increased pressure of several bad years of recruitment caused by the red noise autocorrelation [22]. A sequence of weak year classes may, combined with fishing, lead to a severe reduction in stock size and make the stock more vulnerable to collapse. Increased vulnerability due to slower growth rate has been found in tuna species/populations [58].

By analysing a broad range of parameter values, we can generalize that small-sized and, in particular, slow-growing species are more susceptible to environmental variability. This variability in the recruitment propagates throughout the stock and is detectable in the spawning stock biomass (electronic supplementary material, figure S3). Thus, management for small-sized and slow-growing species may involve adjusting harvesting in order to counterbalance bad years of recruitment. Finally, as previously reported (e.g. [1–4]), we find that a healthy age structure may buffer environmental effects and increase the resilience to disturbances at the stock level.

Ethics. *Research Ethics*. We were not required to complete an ethical assessment prior to conducting this research. *Animal ethics*. We were not required to complete an ethical assessment prior to conducting our research.
Data accessibility. Data for the five fish families in the figures and for the five species are available from the 'rfishbase' package and the *FishBase* database. The R-code for the model is provided in electronic supplementary material.
Permission to carry out fieldwork. No permissions were required prior to conducting our research.
Authors' contributions. L.F. participated in the design and set-up of the study, ran the model and wrote the manuscript. R.v.G. participated in the design of the study and helped to set up the model and draft the manuscript. J.M.D. and Ø.L. participated in the design of the study and helped draft the manuscript. K.H.A. designed and coordinated the study and helped draft the manuscript. All authors gave final approval for publication.
Competing interests. The authors declare no competing interests.
Funding. L.F. and R.v.G. were funded by the MARmaED project, which has received funding from the European Union's Horizon 2020 and Innovation programme under the Marie Skłodowska-Curie grant no. 675997. R.v.G. was also funded through the European Maritime Fisheries Fund (EMFF) of the EU and the State of Mecklenburg-Vorpommern (Germany) (grant no. MV-I.18-LM-004, B 730117000069). The results of this publication reflect only the authors' views, and the European Commission is not responsible for any use that may be made of the information it contains. J.M.D. and Ø.L. were supported by the Research Council of Norway through the projects SPACESHIFT (280468) and FISHCOM (280467), respectively. K.H.A. was supported by The Centre for Ocean Life, a Villum Kahn Rasmussen Centre of Excellence funded by the Villum Foundation.

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
