## [Reviewer comments · Royal Society Open Science]

Review History

RSOS-190063.R0 (Original submission)

Review form: Reviewer 1

Is the manuscript scientifically sound in its present form?

Yes

Are the interpretations and conclusions justified by the results?

No

Is the language acceptable?

Yes

Is it clear how to access all supporting data?

Yes

Do you have any ethical concerns with this paper?

No

Have you any concerns about statistical analyses in this paper?

No

Recommendation?

Major revision is needed (please make suggestions in comments)

Comments to the Author(s)

The authors used an age-structured and trait-based model to explore the impacts of environmental variations and fishing on the dynamics and the sensitivity of fish stocks. They estimated the vital rates of stocks and applied red or white environmental noise and low or high fishing mortality to the stocks. They found that asymptotic size and growth parameters have the highest impacts on the variability of stocks and fast growing and large-sized fish stocks tend to be less vulnerable to disturbances. The paper is pretty well written and follows a good logic despite of occasional grammatical errors. The study design and methods used are sound and scientific questions are important. Some of the results are novel and may have significant management/policy implications. However, we believe there are issues the authors need to address before the paper can be accepted for publications. We divide the comments into "General comments" and "Specific comments". We recommend that the paper be accepted after major revisions.

General comments

1. One of the most important issues in a simulations study design is biological realisms. Although the simulation study did use biological parameters of fish species the authors selected in this study, the choices of variability levels were not well justified. More biological justifications on authors' choices of parameters and levels of uncertainty are needed.
2. The authors estimated and evaluated several vital rates of fish stocks and environmental variability under different levels of fishing mortality. However, the authors have not looked into potential variability in growth among individuals. For a given species, if there are a large variabilities among individuals, this may lead to large variability in age at recruitment, resulting in fishery recruitment consisting of multiple year classes. This type of variability may be very important in studying robustness of fish stocks with respect to fishing mortality and environmental variability. Some relevant analyses would be appreciated by the readers.
3. Some parts of the manuscript is hard to follow. Technical details could be better organized and stated to make the presentation more logical.
4. The authors could further discuss the sources and justifications of environmental noise and forms they take. They could be temperature, hydro character, salinity or any other abiotic factors. Again, it would be good to justify the biological realisms for the choices of parameters and variability. Possible changes in environments may also influence the growth and recruitment dynamics, and it is unclear if such a connection is considered in the simulation when the environmental variations are introduced.

Specific comments

1. We would like to see more specific conclusions regarding the influence of different environmental noises on the stocks with different life history characteristics.
2. All the equations should be numbered in order in the manuscript.
3. The authors should state explicitly that only species within the five families are analyzed in this study. Without such information, the simulated parameters and age class would be suspicious at the first glance.
4. The authors used a combination of independently generated ∞ and A to simulate fishes with difference life history. Such an approach may introduce unrealistic life history patterns. These life history parameters tend to be correlated with other. Thus, sampling these parameters from a joint-distribution may make more biological sense. Additionally, to what extent are extreme combinations reliable (like high ∞ with high A, or low ∞ with low A)?
5. The authors used 100 age classes regardless of species. Would this cause bias in the size of SSB for species with a short longevity? Particularly, imagine a fish with low ∞ and high A, it is unlike

to have longevity of 100. However, its high productivity would cause considerable bias in this case.

6. No plus-group for maximum age is mentioned in the matrix in line 110. If the plus-group is not considered, it would be another source of bias with no fishing mortality scenarios.

7. The authors reduce the abundance of fast-growing species with short life cycles. It is necessary to state what is the criterion for "fast growing species with short life cycles". This makes a joint-distribution of life-history more plausible in this case.

8. The author parameterized the framework with some arbitrarily designated values without considering their uncertainty. Would this diminish the biological realism of the simulation.

9. Line 60: "substituted" should be "compensated".

10. Line 201: "Fishing also reduces the numbers at age in our model, leading to shorter-lived species." What does this mean?

Review form: Reviewer 2

Is the manuscript scientifically sound in its present form?

Yes

Are the interpretations and conclusions justified by the results?

No

Is the language acceptable?

Yes

Is it clear how to access all supporting data?

Yes

Do you have any ethical concerns with this paper?

No

Have you any concerns about statistical analyses in this paper?

No

Recommendation?

Reject

Comments to the Author(s)

Please see attached file (Appendix A).

Decision letter (RSOS-190063.R0)

27-Feb-2019

Dear Ms Färber:

Manuscript ID RSOS-190063 entitled "Population variability under stressors depending on growth and asymptotic size" which you submitted to Royal Society Open Science, has been reviewed. The comments from reviewers are included at the bottom of this letter.

In view of the criticisms of the reviewers, the manuscript has been rejected in its current form. However, a new manuscript may be submitted which takes into consideration these comments.

Please note that resubmitting your manuscript does not guarantee eventual acceptance, and that your resubmission will be subject to peer review before a decision is made.

Your resubmitted manuscript should be submitted by 27-Aug-2019. If you are unable to submit by this date please contact the Editorial Office.

on behalf of Dr Punidan Jeyasingh (Associate Editor) and Professor Kevin Padian (Subject Editor)
openscience@royalsociety.org

Subject Editor Comments to Author:

Thank you for your interesting submission. I would like to give you the chance to rework it and edit it with more time than our "major revision" timetable provides. Additionally, the reviewers raise serious concerns that you should have sufficient time to address. Finally, the English is very good but there are still numerous errors, and it needs careful editing by a native speaker of English. Also, please do not right-justify the manuscript. Please attend to the reviewers' comments carefully, and we look forward to a re-submission.

Associate Editor Comments to Author (Dr Punidan Jeyasingh):

This is a well-written manuscript exploring the drivers of fish population variability using a modeling approach. The manuscript was reviewed by two experts. They raise several key points (ranging from better literature review to rationalizing the design) that need to be addressed. I felt the reviews were fair and constructive. With much gratitude to the expert reviewers, I invite the authors to revise their manuscript.

Reviewers' Comments to Author:

Reviewer: 1

Comments to the Author(s)

The authors used an age-structured and trait-based model to explore the impacts of environmental variations and fishing on the dynamics and the sensitivity of fish stocks. They estimated the vital rates of stocks and applied red or white environmental noise and low or high fishing mortality to the stocks. They found that asymptotic size and growth parameters have the highest impacts on the variability of stocks and fast growing and large-sized fish stocks tend to be

less vulnerable to disturbances. The paper is pretty well written and follows a good logic despite of occasional grammatical errors. The study design and methods used are sound and scientific questions are important. Some of the results are novel and may have significant management/policy implications. However, we believe there are issues the authors need to address before the paper can be accepted for publications. We divide the comments into “General comments” and “Specific comments”. We recommend that the paper be accepted after major revisions.

General comments

1. One of the most important issues in a simulations study design is biological realism. Although the simulation study did use biological parameters of fish species the authors selected in this study, the choices of variability levels were not well justified. More biological justifications on authors' choices of parameters and levels of uncertainty are needed.
2. The authors estimated and evaluated several vital rates of fish stocks and environmental variability under different levels of fishing mortality. However, the authors have not looked into potential variability in growth among individuals. For a given species, if there are large variabilities among individuals, this may lead to large variability in age at recruitment, resulting in fishery recruitment consisting of multiple year classes. This type of variability may be very important in studying robustness of fish stocks with respect to fishing mortality and environmental variability. Some relevant analyses would be appreciated by the readers.
3. Some parts of the manuscript is hard to follow. Technical details could be better organized and stated to make the presentation more logical.
4. The authors could further discuss the sources and justifications of environmental noise and forms they take. They could be temperature, hydro character, salinity or any other abiotic factors. Again, it would be good to justify the biological realism for the choices of parameters and variability. Possible changes in environments may also influence the growth and recruitment dynamics, and it is unclear if such a connection is considered in the simulation when the environmental variations are introduced.

Specific comments

1. We would like to see more specific conclusions regarding the influence of different environmental noises on the stocks with different life history characteristics.
2. All the equations should be numbered in order in the manuscript.
3. The authors should state explicitly that only species within the five families are analyzed in this study. Without such information, the simulated parameters and age class would be suspicious at the first glance.
4. The authors used a combination of independently generated ∞ and A to simulate fishes with difference life history. Such an approach may introduce unrealistic life history patterns. These life history parameters tend to be correlated with other. Thus, sampling these parameters from a joint-distribution may make more biological sense. Additionally, to what extent are extreme combinations reliable (like high ∞ with high A, or low ∞ with low A)?
5. The authors used 100 age classes regardless of species. Would this cause bias in the size of SSB for species with a short longevity? Particularly, imagine a fish with low ∞ and high A, it is unlike to have longevity of 100. However, its high productivity would cause considerable bias in this case.
6. No plus-group for maximum age is mentioned in the matrix in line 110. If the plus-group is not considered, it would be another source of bias with no fishing mortality scenarios.
7. The authors reduce the abundance of fast-growing species with short life cycles. It is necessary to state what is the criterion for “fast growing species with short life cycles”. This makes a joint-distribution of life-history more plausible in this case.
8. The author parameterized the framework with some arbitrarily designated values without considering their uncertainty. Would this diminish the biological realism of the simulation.
9. Line 60: “substituted” should be “compensated”.

10. Line 201: "Fishing also reduces the numbers at age in our model, leading to shorter-lived species." What does this mean?

Reviewer: 2

Comments to the Author(s)
Please see attached file.

Author's Response to Decision Letter for (RSOS-190063.R0)

See Appendix B.

RSOS-192011.R0

Review form: Reviewer 3

Is the manuscript scientifically sound in its present form?

Yes

Are the interpretations and conclusions justified by the results?

Yes

Is the language acceptable?

Yes

Do you have any ethical concerns with this paper?

No

Have you any concerns about statistical analyses in this paper?

No

Recommendation?

Accept with minor revision (please list in comments)

Comments to the Author(s)

In general, I found the author responses thorough and well motivated. Therefore, I recommend a minor revision addressing the minor errors I found in the paper.

The only major thing that would have been interesting to add to the current manuscript is an analysis showing the effect of noise on other processes in the model (even though recruitment may be the most uncertain process in the model, other parts of the recruitment may equally likely be affected (like through R_{max} or a direct additive effect on R_t)). This would probably have major implications for responses with respect to the different noise processes (white and red noise). However, I think the authors have already done a thorough job in addressing all reviewer comments and therefore accept if no such analysis is conducted.

Here are some minor comments:

- * Equation numbers should be corrected.
- * There is a time index missing in the second part of eq.(2).
- * Inequalities with wrong direction (l. 204 & l. 205)
- * (Eq.(2) in table 1) wrongly specified model of red noise. It should read: $x_{red}(t) = c * x_{red}(t-1) + x_{white}(t) * \sqrt{1-c^2}$
- * Please state in the figure legend to figure 2 the response variable being used. Is it SSB or recruitment?

Review form: Reviewer 4

Is the manuscript scientifically sound in its present form?

Yes

Are the interpretations and conclusions justified by the results?

Yes

Is the language acceptable?

No

Do you have any ethical concerns with this paper?

No

Have you any concerns about statistical analyses in this paper?

No

Recommendation?

Major revision is needed (please make suggestions in comments)

Comments to the Author(s)

I found the paper difficult to follow. One reason for this is that a lot of the sentences are long and unclear. This may often be language issue that can relatively easily be solved, but may also be due to unclear thinking.

Details

- In the title, abstract and introduction you use the terms growth (and growth rate) and asymptotic size. It wasn't until well into the methods that I became confident that it is body mass growth and asymptotic body mass (size) that was meant. Make this clear already in the title, abstract and introduction. (The way the expressions are used, you could have be referring to population growth and a parameter for some asymptotic population size).
- Abstract line 32-35. Long awkward sentence. Rewrite.
- Line 37-38. I find it strange to claim that "Our study indicates that the variability of stock recruitment and biomass can be driven by the stock's asymptotic size and growth rate". The body mass growth parameters you focus on do not drive variability, but may be associated with patterns in the population variability driven by environmental variability.
- Line 38-40. Fine, but cut line 37-40 into at least two sentences.
- Line 40-41. Is this a general result? If so make it clear that it is independent of body mass growth parameter values.

- Line 41-43. Delete? Seems to be repetition + I am not convinced by the claim that your focal body growth parameters are “crucial for a species resilience toward s disturbances”. Reword: “expected to be important predictors of” rather than “crucial for” maybe?
- Make clear what is the new knowledge gained from the work.

Introduction

- Line 57-91. I find most of the first part of the introduction difficult to read. I can guess what the authors want to say, but the text is almost incomprehensible. Cut the text into more and simpler sentences and check that the words used are appropriate. The information should lead up to some conclusion. This part need a proper job.
- E.g. I believe the second sentence can be deleted (line 57-60). The third sentence could be cut down to “The early life stages of fish are particularly sensitive to environmental variation, and this sensitivity can result in large variability in the size of the spawning stock and the stock available for the fisheries.”
- It is not clear what new knowledge we gain from the study. The last few sentence do not make this clear even though the authors claim they do so in the response to reviewer II. Be direct. E.g. Previous studies have shown....Our study expand on this by showing that....

Methods

- Line 119-120. Delete sentence.
- Line 120-129. The text need some work.
- Table 2. Include a definition of the index i.
- Line 140. The age index is wrong in for the abundance the next year $N_{i,t+1}$ should be $N_{i+1,t+1}$
- Line 151-152. Poor sentence.
- Delete eq 7 and 8 in the main text. Refere to the equations in the Table 1.
- Use the same expressions for the parameters throughout. Eg. In table 2 A is called “Growth costant” while in most of the manuscript you call it growth or growth rate. I suggest “body mass growth rate”
- Line 181-183. Unclear sentence. Also, what did you do when populations crashed.
- Table 3. Include common names in addition to latin species and family names. I did not find coloumn 3 and 5 particularely interesting.

Results

- Line 204-205. Seems like > in brackets should be <.
- Line 213-216. Long sentence. Rewrite.
- Line 218. Reword “disturbances” “variability”?
- Line 221-224. Long sentence.

Discussion

- Line 230-234. Long winded sentence. Also the 20% variability must depend on the variance in the environment used in the simulations.
- Line 245. Reword “fewer”
- Line 247-249. You haven’t shown a reduction in the number of cohorts in the populations associated with fishing.
- Line 250-251. Unclear sentence.
- Line 263-265. Unclear sentence.
- Line 266-269. Long unclear sentence. It is unclear what is eroded.
- Line 269-270. Poor sentence. The term overfishing imply sensitivity to fishing.
- Line 270-271. Poor sentence. (change “the decreased biomass” to “a decreased biomass”?)
- Line 273. Not “inhabited”, wrong word.
- Line 274-277. Poor sentence. What is an “older population structure” ?
- Line 282. Reword “sustain”
- Line 287. Reword “on” to “about”?
- Line 292-296. Poor sentence.
- Line 306-309. Poor sentence.
- Line 311-313. Long poor sentence. Delete?
- Line 317-319. Long poor sentence.
- Line 327-329. Poor sentence.

- Legend to figure 2. Needs tightening up. The parameters for the species are already given in table 3 and do not need to be repeated (refer to table 3).
- Legend to figure 3. The parameters for the species are already given in table 3 and do not need to be repeated (refer to table 3).

Decision letter (RSOS-192011.R0)

19-Dec-2019

Dear Dr Färber,

The Subject Editor assigned to your paper ("Population variability under stressors depending on growth and asymptotic size") has now received comments from reviewers. We would like you to revise your paper in accordance with the referee and Associate Editor suggestions which can be found below (not including confidential reports to the Editor). Please note this decision does not guarantee eventual acceptance.

Please submit a copy of your revised paper before 11-Jan-2020. Please note that the revision deadline will expire at 00.00am on this date. If we do not hear from you within this time then it will be assumed that the paper has been withdrawn. In exceptional circumstances, extensions may be possible if agreed with the Editorial Office in advance. We do not allow multiple rounds of revision so we urge you to make every effort to fully address all of the comments at this stage. If deemed necessary by the Editors, your manuscript will be sent back to one or more of the original reviewers for assessment. If the original reviewers are not available we may invite new reviewers.

When submitting your revised manuscript, you must respond to the comments made by the referees and upload a file "Response to Referees" in "Section 6 - File Upload". Please use this to document how you have responded to each of the comments, and the adjustments you have made. In order to expedite the processing of the revised manuscript, please be as specific as possible in your response.

- Ethics statement

- Data accessibility

It is a condition of publication that all supporting data are made available either as supplementary information or preferably in a suitable permanent repository. The data accessibility section should state where the article's supporting data can be accessed. This section should also include details, where possible of where to access other relevant research materials

such as statistical tools, protocols, software etc can be accessed. If the data has been deposited in an external repository this section should list the database, accession number and link to the DOI for all data from the article that has been made publicly available. Data sets that have been deposited in an external repository and have a DOI should also be appropriately cited in the manuscript and included in the reference list.

If you wish to submit your supporting data or code to Dryad (<http://datadryad.org/>), or modify your current submission to dryad, please use the following link:
<http://datadryad.org/submit?journalID=RSOS&manu=RSOS-192011>

- **Competing interests**

- **Authors' contributions**

- **Acknowledgements**

- **Funding statement**

on behalf of Dr Punidan Jeyasingh (Associate Editor) and Kevin Padian (Subject Editor)
openscience@royalsociety.org

Editorial Comments to Author:

As you have been requested to edit the written English, you must provide proof that you have done so: acceptable proof includes a certificate of language-editing from a language editing

service or a signed letter from a native speaker of English. If you do not provide this proof, your manuscript may be returned to you."

For information about language editing services endorsed by the Royal Society, please follow the link below:

<https://royalsociety.org/journals/authors/language-polishing/>

Associate Editor Comments to Author (Dr Punidan Jeyasingh):

I thank the authors for a thorough revision. This version is much improved, and was assessed by two experts. While the reviews were more favorable this time around, both reviewers felt that the language and presentation could be further improved. The reviewers make a number of recommendations to mitigate the issue. I felt the reviews were fair and constructive. I recommend the authors make these revisions and send the manuscript to an anglophone (ideally one who isn't an expert in the field) for comments to improve comprehensibility before submitting another version. The science is strong, however, the work needs to be presented in a clearer fashion. I look forward to reading a revised version.

Reviewer comments to Author:

Reviewer: 3

Comments to the Author(s)

In general, I found the author responses thorough and well motivated. Therefore, I recommend a minor revision addressing the minor errors I found in the paper.

The only major thing that would have been interesting to add to the current manuscript is an analysis showing the effect of noise on other processes in the model (even though recruitment may be the most uncertain process in the model, other parts of the recruitment may equally likely be affected (like through R_{max} or a direct additive effect on R_t)). This would probably have major implications for responses with respect to the different noise processes (white and red noise). However, I think the authors have already done a thorough job in addressing all reviewer comments and therefore accept if no such analysis is conducted.

Here are some minor comments:

- * Equation numbers should be corrected.
- * There is a time index missing in the second part of eq.(2).
- * Inequalities with wrong direction (l. 204 & l. 205)
- * (Eq.(2) in table 1) wrongly specified model of red noise. It should read: $x_{red}(t) = c * x_{red}(t-1) + x_{white}(t) * \sqrt{1-c^2}$
- * Please state in the figure legend to figure 2 the response variable being used. Is it SSB or recruitment?

Reviewer: 4

Comments to the Author(s)

I found the paper difficult to follow. One reason for this is that a lot of the sentences are long and unclear. This may often be language issue that can relatively easily be solved, but may also be due to unclear thinking.

Details

- In the title, abstract and introduction you use the terms growth (and growth rate) and asymptotic size. It wasn't until well into the methods that I became confident that it is body mass growth and asymptotic body mass (size) that was meant. Make this clear already in the title, abstract and introduction. (The way the expressions are used, you could have been referring to population growth and a parameter for some asymptotic population size).
- Abstract line 32-35. Long awkward sentence. Rewrite.
- Line 37-38. I find it strange to claim that "Our study indicates that the variability of stock recruitment and biomass can be driven by the stock's asymptotic size and growth rate". The body mass growth parameters you focus on do not drive variability, but may be associated with patterns in the population variability driven by environmental variability.
- Line 38-40. Fine, but cut line 37-40 into at least two sentences.
- Line 40-41. Is this a general result? If so make it clear that it is independent of body mass growth parameter values.
- Line 41-43. Delete? Seems to be repetition + I am not convinced by the claim that your focal body growth parameters are "crucial for a species resilience toward s disturbances". Reword: "expected to be important predictors of" rather than "crucial for" maybe?
- Make clear what is the new knowledge gained from the work.

Introduction

- Line 57-91. I find most of the first part of the introduction difficult to read. I can guess what the authors want to say, but the text is almost incomprehensible. Cut the text into more and simpler sentences and check that the words used are appropriate. The information should lead up to some conclusion. This part needs a proper job.
- E.g. I believe the second sentence can be deleted (line 57-60). The third sentence could be cut down to "The early life stages of fish are particularly sensitive to environmental variation, and this sensitivity can result in large variability in the size of the spawning stock and the stock available for the fisheries."
- It is not clear what new knowledge we gain from the study. The last few sentences do not make this clear even though the authors claim they do so in the response to reviewer II. Be direct. E.g. Previous studies have shown....Our study expands on this by showing that....

Methods

- Line 119-120. Delete sentence.
- Line 120-129. The text needs some work.
- Table 2. Include a definition of the index i .
- Line 140. The age index is wrong in for the abundance the next year $N_{i,t+1}$ should be $N_{i+1,t+1}$
- Line 151-152. Poor sentence.
- Delete eq 7 and 8 in the main text. Refer to the equations in the Table 1.
- Use the same expressions for the parameters throughout. Eg. In table 2 A is called "Growth constant" while in most of the manuscript you call it growth or growth rate. I suggest "body mass growth rate"
- Line 181-183. Unclear sentence. Also, what did you do when populations crashed.
- Table 3. Include common names in addition to latin species and family names. I did not find column 3 and 5 particularly interesting.

Results

- Line 204-205. Seems like $>$ in brackets should be $<$.
- Line 213-216. Long sentence. Rewrite.
- Line 218. Reword "disturbances" "variability"?
- Line 221-224. Long sentence.

Discussion

- Line 230-234. Long winding sentence. Also the 20% variability must depend on the variance in the environment used in the simulations.
- Line 245. Reword "fewer"
- Line 247-249. You haven't shown a reduction in the number of cohorts in the populations associated with fishing.
- Line 250-251. Unclear sentence.
- Line 263-265. Unclear sentence.

- Line 266-269. Long unclear sentence. It is unclear what is eroded.
- Line 269-270. Poor sentence. The term overfishing imply sensitivity to fishing.
- Line 270-271. Poor sentence. (change "the decreased biomass" to "a decreased biomass"?)
- Line 273. Not "inhabited", wrong word.
- Line 274-277. Poor sentence. What is an "older population structure" ?
- Line 282. Reword "sustain"
- Line 287. Reword "on" to "about"?
- Line 292-296. Poor sentence.
- Line 306-309. Poor sentence.
- Line 311-313. Long poor sentence. Delete?
- Line 317-319. Long poor sentence.
- Line 327-329. Poor sentence.
- Legend to figure 2. Needs tightening up. The parameters for the species are already given in table 3 and do not need to be repeated (refer to table 3).
- Legend to figure 3. The parameters for the species are already given in table 3 and do not need to be repeated (refer to table 3).

Author's Response to Decision Letter for (RSOS-192011.R0)

See Appendix C.

Decision letter (RSOS-192011.R1)

03-Feb-2020

Dear Dr Durant,

It is a pleasure to accept your manuscript entitled "Population variability under stressors is dependent on body mass growth and asymptotic body size." in its current form for publication in Royal Society Open Science. The comments of the reviewer(s) who reviewed your manuscript are included at the foot of this letter.

Kind regards,

Anita Kristiansen
Editorial Coordinator

on behalf of Dr Punidan Jeyasingh (Associate Editor) and Kevin Padian (Subject Editor)
openscience@royalsociety.org

Associate Editor Comments to Author (Dr Punidan Jeyasingh):

I thank the authors for a thorough revision. This version is much improved and in my opinion, ready for press. With much gratitude to the expert reviewers, I congratulate the authors for this contribution.

Appendix A

Review of Ms #RSOS-190063, Färber et al. to Royal Society Open Science

Färber et al. use an age-structured model of a fish population, accounting for how its dynamics vary with body size, to analyze the sensitivity of the population's vital rates to environmental variability with different degree of autocorrelation (white/red noise) and harvesting (low/high). They do so by using a model published by Andersen & Beyer (2015) and incorporate environmental variability, and slightly reparameterizing the model. They calculate variability over 499 time steps for a range of combinations of asymptotic size and growth rates. They also chose five marine fish species (which differ to varying degree in their asymptotic size and growth rates), and subject these to two levels of fishing, combined with the two types of noise, to study the effect on recruitment. They conclude that the least variable fish stocks are those with large asymptotic sizes and at the same time fast growing life histories, and that fishing increases population variability.

The study addresses a common and well-studied aspect of population dynamics: how populations respond dynamically to environmental variation, how it propagates to later life stages than the one it directly acts on, and the interactive effects of mortality on these later stages. However, the introduction fails to recognize the previous literature in this field (its vast, some suggestions on literature are included below). Without a sufficient background on earlier studies of environmental noise of different color and how it together with mortality shapes population dynamics, it is hard to see which the current knowledge gap is, and how the current study may fill that. For example, previous studies have already addressed the interactive effect of fishing and noise colour (acting on recruitment, as in this study) (e.g., Kuparinen et al 2014). So, it is unclear where the novelty lies in the work by Färber et al, and the authors fails to convincingly demonstrate that in their introduction.

It may be that the role of growth and body size for population dynamical responses to noise color and mortality has not been previously demonstrated. If so, I urge the authors to carefully review and synthesize available studies on noise color and population dynamics to demonstrate this. And, to relate to these and in your revised discussion clearly show how your study complements them.

The presentation of methods and results is also somewhat unclear, making it hard to follow the set-up and part of the findings. The method section lacks a description of which analyses that were made using the model. From the results text and figure 3 (legend vs axis label) it is unclear whether it is both the level and the variation of recruitment that was studied. The authors also make conclusions on mechanisms underlying the observed patterns (e.g. that there is a buffering effect of the large body size) without supporting these by any analyses. Similarly, they also suggest that their results may not be robust to the setup of their noise effect (that it acts on recruitment), but do not test this in their model (i.e. adding noise to other population processes).

Some suggestions on studies in the field of noisy environment and population dynamics:

Fowler, M. S. & Ruokolainen, L. Colonization, covariance and colour: environmental and ecological drivers of diversity–stability relationships. *J. Theor. Biol.* 324, 32–41 (2013).

Halley, J. M. Ecology, evolution and $1/f$ -noise. *Trends Ecol. Evol.* 11, 33–37 (1996).

Kuparinen, A., Keith, D. M. & Hutchings, J. A. Increased environmentally driven recruitment variability decreases resilience to fishing and increases uncertainty of recovery. *ICES J. Mar. Sci.* 71, 1507–1514 (2014).

Ruokolainen, L., Ranta, E., Kaitala, V. & Fowler, M. S. Community stability under different correlation structures of species' environmental responses. *J. Theor. Biol.* 261, 379–387 (2009).

Ruokolainen, L., Linden, A., Kaitala, V. & Fowler, M. S. Ecological and evolutionary dynamics under coloured environmental variation. *Trends Ecol. Evol.* 24, 555–563 (2009).

Minor comments:

L24: “fluctuations in recruitment, which propagates throughout harvested biomass” sounds strange, as the harvested biomass is the killed biomass. So either change to “harvestable biomass” or “propagates to harvested biomass”

L 92 & tables; suggest you switch order of Table 1 & 2, as the parameters (Table 1) are only interpretable in the light of the equations (Table 2)

L95 life table is not a method for time integration, suggest you write matrix model instead.

L104 suggest you remove “(freshwater or marine)”; this classification is irrelevant as origin is irrelevant for your question and model set-up

L110: these are usually written with a vector of abundances multiplied by a projection matrix

L142: remove excess “the”

L142: state the date when you downloaded data from Fishbase

L142: what parameter space did you obtain from Fishbase (for which parameters)? And what did you plot in there and why? Are you referring to the species-specific values of growth rates, asymptotic weights and maturation ages? If so, clarify that in the first sentence bringing this up.

L143-144: “data was plotted as a convex hull”; unclear why (and which) data was plotted like this.

144 smallest convex area that contains all the data points)

L147-148: “most harvested species” change to “most intensely harvested species”

L160 ff here I'm missing some explanations of which analyses you made in the model

L164 to support the conclusion that the recruitment variability imposed by the noisy mortality is detectable also in SSB this needs to be shown (in appendix figs) for all species

L174 rephrase to “Herring showed larger recruitment variability in response to environmental noise than the other five...”; please add figure reference for this statement

L177-L183 if Golden redfish and Atlantic cod have similar life histories, what is the explanation to the differences in how they respond to fishing? But, judging from fig 2, Golden redfish is about 10 times smaller than cod and has about 5 times slower growth rate. Isn't it? If so, could that explain their different responses.

L186-188 From figure A1 it seems as if the panels with red noise have consistently darker color (higher CV of recruitment) than those with white noise, at least for the cases with fishing. So didn't it have an effect on population variability? Maybe use a different color scheme for your CV-levels would make it easier to see?

L226-229 this motivation should be presented already in the methods (and, preferably also in the introduction)

L229-231 yes, previous studies have shown that in which life stage the environmental noise is included has strong effects on the outcome. Do your conclusions hold if the noise is added to another life-stage process? Show (by supplementary analyses) or motivate.

L232 please expand and clarify what these could be, based on previous studies. Alternatively, include these analyses in the supplement.

L237-239 you do not have freshwater/marine water in your model, so what's the explanation in your model? Link the smaller size (and their growth rate) to what red noise does to the dynamics.

L245-247 didn't you test whether that statement holds? You did runs with higher recruitment efficiency in fig A2. Suggest you refer to that here.

L429 table legend hard to understand. Isn't it just model equations? If so, I suggest you rephrase to Model equations.

Figure 3: if this is depicting recruitment variability, what does the y-axis label mean? Isn't this a proportion (R/R_{max})? So isn't the variability shown as the range of the boxplot increases? In that case, the legend needs to be corrected (now reads "boxplots of recruitment variability", which means that the boxplots then depict the variation in that variability). If the legend is correct, then the y-axis labels need to be corrected.

Appendix B

Dear Editor,

Thank you for the possibility of reworking our submission, even with an extension of the deadline. We have made extensive changes in order to address all the concerns in the useful comments from the reviewers. We included more literature to highlight where our research fits into these existing studies and we improved the reasoning for our study. Especially, we addressed the concern by Reviewer 1 and conducted a sensitivity analysis of our parameters, in combination with better biological justification of our parameter choice (Appendix Figure A5). The sensitivity analysis showed that the exact choice of the parameter value is not important for the main results of our study. Furthermore, we clarified concerns from the reviewers with additional Figures in the Appendix of the manuscript.

Additionally, we reworked the manuscript for mistakes in the language and we hope to have improved its readability.

Detailed Response to the Reviewers

Reviewer I

The authors used an age-structured and trait-based model to explore the impacts of environmental variations and fishing on the dynamics and the sensitivity of fish stocks. They estimated the vital rates of stocks and applied red or white environmental noise and low or high fishing mortality to the stocks. They found that asymptotic size and growth parameters have the highest impacts on the variability of stocks and fast growing and large-sized fish stocks tend to be less vulnerable to disturbances. The paper is pretty well written and follows a good logic despite of occasional grammatical errors. The study design and methods used are sound and scientific questions are important. Some of the results are novel and may have significant management/policy implications. However, we believe there are issues the authors need to address before the paper can be accepted for publications. We divide the comments into “General comments” and “Specific comments”. We recommend that the paper be accepted after major revisions.

Thank you for the positive feedback on our manuscript, and for identifying areas in need of more work and analyses. We addressed all the issues raised by an extensive new sensitivity analysis and made changes in the manuscript where appropriate.

- 1. One of the most important issues in a simulations study design is biological realisms. Although the simulation study did use biological parameters of fish species the authors selected in this study, the choices of variability levels were not well justified. More biological justifications on authors' choices of parameters and levels of uncertainty are needed.**

We acknowledge the concern of Reviewer #1. The parameter values we chose for this simulation study are taken from literature data (see e.g. also Andersen and Beyer, 2015, Kokkalis et al., 2016). The variability was the one inherent to the environmental noise (red or white) chosen. We acknowledge that our results may depend on our definition of the environmental noise (see method part, Table 1 Eq. 11 and 12). To ascertain our results we have conducted a sensitivity analysis to the exact choice of parameter values. Furthermore, we have added detailed justification for the choice of the values of the biological parameters. These are now given in the Appendix, visualized in Figure A5 and reiterated below.

- *Metabolic scaling factor n : $n = \frac{3}{4}$ (West et al., 1997) because it acknowledges that the surface scales with weight $> \frac{2}{3}$ since it may be fractal (Andersen, 2019; Andersen and Beyer, 2015)*
- *Physiological mortality α : Based on calculations from (Gislason et al., 2010) on natural mortality of fish and growth rate (Andersen, 2019; Andersen and Beyer, 2015).*
- *Start of fishing η_F : In many fish species, already juveniles are vulnerable to fishing gear (see e.g. (Dickey-Collas et al., 2010; Hysten et al., 2008) as example for Atlantic cod and Atlantic herring) and thus we decided to have an entry level slightly before the maturity at $W_\infty 0.25$.*
- *Width of trawl-selectivity u_F : Our own choice, choosing a not too hard knife-edge transition, to allow for gradual shift in selectivity, but see sensitivity analysis (Fig. A5)*
- *Size at maturity η_m : From asymptotic length to length at maturity ratios (Beverton, 1992) one gets ranges from around 0.4 to 0.8. When converted to weight as used here with $w = hl^3$, where h is a coefficient in the weight length-relationship with $h=0.01$ (Froese, 2006) we can approximate the value of $W_\infty 0.25$.*
- *Width of maturity switching function u_m : Our own choice, smoother than for the fishing switching function. See sensitivity analysis (Fig. A5)*
- *Recruitment efficiency ε_r : A value for recruitment efficiency is difficult to find in literature, we used an estimation from (Hartvig et al., 2011), which is based on calculations from n bioenergetics model and empirical measurements*
- *Egg weight w_{egg} : From data from FishBase (Froese and Pauly, 2018) one can see that the geometric mean of teleosts egg weight stays consistently similar over the weight range and is around 1 mg (see e.g. Figure 8.2 in (Andersen, 2019)).*
- *Fraction of energy used for activity ε_a : Value derived from data from (Gunderson, 1997) and see Fig. 2a (Andersen and Beyer, 2015).*
- *White noise standard deviation σ : Our own choice, to see large enough variability, but see sensitivity analysis and compare Botsford et al. 2014*
- *Red noise correlation coefficient c : Our own choice, however, in order to mimic large scale environmental drivers such as the NAO, we set the correlation high,*

here to 0.9, corresponding to a periodicity of around 10 years. But see also sensitivity analysis.

We include now a paragraph of model limitation in our discussion.

The authors estimated and evaluated several vital rates of fish stocks and environmental variability under different levels of fishing mortality. However, the authors have not looked into potential variability in growth among individuals. For a given species, if there are a large variabilities among individuals, this may lead to large variability in age at recruitment, resulting in fishery recruitment consisting of multiple year classes. This type of variability may be very important in studying robustness of fish stocks with respect to fishing mortality and environmental variability. Some relevant analyses would be appreciated by the readers.

In our study, we did not explicitly account for the within-species (individual) differences. We consider this beyond the scope of the paper. However, we acknowledge that this is a limitation of our approach, and we have now added a discussion on this topic in the manuscript. It reads:

“In this study, we used a simplified model, to obtain general results (cf. Levins, 1966). However, this means that the model has some limitations, e.g., the model does not account for individual variations in growth nor does the model account for environmental impact on growth. Despite these limitations, we argue that our modelling approach is useful to obtain information on general patterns across fish life-histories, but we acknowledge that not all parameters used in the model fit to individual species or populations. Also, note that our sensitivity analyses (Appendix Figure A5) highlights how the exact values of the parameters do not change the general the results of our study, however if changes occur they are stronger observed in the slow growing and small size classes, especially in the scenarios with fishing. These populations are already responding strongly to environmental variability and fishing pressure.” (L313 ff.)

Some parts of the manuscript is hard to follow. Technical details could be better organized and stated to make the presentation more logical.

We have rewritten and reorganized the manuscript in depth and hope it is now easier to follow.

- 2. The authors could further discuss the sources and justifications of environmental noise and forms they take. They could be temperature, hydro character, salinity or any other abiotic factors. Again, it would be good to justify the biological realisms for the choices of parameters and variability. Possible changes in environments may also influence the growth and recruitment dynamics, and it is unclear if such a connection is considered in the simulation when the environmental variations are introduced.**

We discuss now further the forms the noise can take (see introduction)

“The environment can have direct effects, for example impacting the growth through changes in temperature, but also indirectly by impacting prey availability (Ottersen et al., 2010). Furthermore, year-to-year variability in water temperature influences productivity in fish stocks (Ottersen et al., 2013).” L63 ff.

However, also state in the discussion that we do not explicitly model this: *“[...] the model does not account for individual variations in growth nor does the model account for environmental impact on growth.”* L314 ff.

Specific comments

- 1. We would like to see more specific conclusions regarding the influence of different environmental noises on the stocks with different life history characteristics.**

“We have added more discussion on this topic in the last paragraph of the discussion, new text include “We see, in our theoretical exploration of two life history parameters, the asymptotic size and growth rate, that size- and growth rate matter for a harvested stock in order to sustain disturbances by environmental noise and fishing pressure. In our model, the noise colour did not lead to strong differences in the response of the stocks, however slightly more stocks collapsed/displayed larger variability under red noise conditions, which could be expected from the literature (e.g. Kuparinen et al., 2014; Lawton, 1988; Ruokolainen et al., 2009; Steele and Henderson, 1984). Especially, slower growing species, in the scenarios with fishing reacted stronger on red noise. This is due to the possible increased pressure of several bad years of recruitment, due to the red noise autocorrelation (Lawton, 1988), leading, next to fishing, to an additional reduction in stock size and make the stock more vulnerable. Increased vulnerability due to slower growth rate was for example also shown in tuna species/populations (Juan-Jordá et al., 2015).” (L308 ff.)
”

- 2. All the equations should be numbered in order in the manuscript.**
We ordered the equations accordingly.
- 3. The authors should state explicitly that only species within the five families are analyzed in this study. Without such information, the simulated parameters and age class would be suspicious at the first glance.**

We highlighted now more that we extracted the five species from the runs as example, however as we tried to be general, we could also use different examples.

- 4. The authors used a combination of independently generated W_∞ and A to simulate fishes with difference life history. Such an approach may introduce unrealistic life history patterns. These life history parameters tend to be correlated with other. Thus, sampling these parameters from a joint-distribution may make more biological sense. Additionally, to what extent are**

extreme combinations reliable (like high W_{∞} with high A, or low W_{∞} with low A)?

We reran the analyses now with the most frequent combination of A and W_{∞} (within the 80 % most common combinations) from the data in Fishbase, where we calculated A from K and W_{∞} from L_{inf} as indicated in the Appendix. We used now data which consists of 80% of the most common species and we set up a logarithmic grid over these most common size/growth combinations and used 40 steps in each direction. For clarifications, we also show now the biological realistic species in the space in the plots, i.e. the area, where Fishbase has data combinations for our A and W_{∞} (black dotted line around the CV values in Fig 2 and A3). And again visualised in Figure A2.

5. The authors used 100 age classes regardless of species. Would this cause bias in the size of SSB for species with a short longevity? Particularly, imagine a fish with low W_{∞} and high A, it is unlike to have longevity of 100. However, its high productivity would cause considerate bias in this case.

Smaller sized and fast growing species have a high instantaneous mortality rate per year, thus their numbers will reduce very quickly and have numbers close to 0 in age classes higher than their natural age. Whereas slow growing and larger species have a slower decrease in their stock numbers and can populate more age classes. In the Figure below you see three examples with a species with fast growth ($A=14 \text{ g}^{1-n} \text{ yr}^{-1}$) and low weight ($W_{\infty}=30 \text{ g}$). Which could for example be close to a Engraulidae; the medium growth ($A=4.8 \text{ g}^{1-n} \text{ yr}^{-1}$) and medium weight ($W_{\infty}=178 \text{ g}$) could be represented by e.g. a European perch; the slow growing ($A=1.8 \text{ g}^{1-n} \text{ yr}^{-1}$) with high weight (ca. 38 kg) could be a lake sturgeon. It can populate > 50 age classes, thus we want to make sure, to have sufficient age classes also for very long-lived species. We thus prefer to keep the 100 age classes to serve as a buffer. We made clarifications in the manuscript, which reads now:

“Each population was divided into a total of $m=100$ age classes, with i indicating any given age. The number of age classes is well above the likely maximum age of most species in the model, however long-lived and slow growing species can reach their asymptotic size at a high age. Thus by setting a high number of age classes, we avoid an involuntary eviction of the old fish in the model. Shorter-lived species will have abundances very close to 0 due to high mortality in early years, so the results will not be biased.” (L155 ff.)

6. No plus-group for maximum age is mentioned in the matrix in line 110. If the plus-group is not considered, it would be another source of bias with no fishing mortality scenarios.

We keep the 100 age classes (see above) so that each age class of a species with slow growth and large weight can grow and contribute to the population, however in the age class of 100, the abundance of all fish is very close to 0 and thus, there is no bias and no plus group needs to be considered.

7. The authors reduce the abundance of fast-growing species with short life cycles. It is necessary to state what is the criterion for “fast growing species with short life cycles”. This makes a joint-distribution of life-history more plausible in this case.

We state now what is a short-life cycle, meaning the fast growing species reach also maturity fast, however, then they will also die, due to high mortality rates (see Figure above to comment 5. And also equation for abundance, Table 1, Eq.2. High mortality due to high growth rates, leads to reduced abundances fast.

8. The author parameterized the framework with some arbitrarily designated values without considering their uncertainty. Would this diminish the biological realism of the simulation.

The parameter values we chose for our simulation study are all taken from literature data and not chosen arbitrary (see e.g. also Andersen and Beyer, 2015; Kokkalis et al., 2016). We have now added information justifying our choice of the biological parameters. These are now given in supplementary material. Note that adding a level of variability to the parameters did not changed our results (see answer to comment 1 and sensitivity analysis Figure A5).

9. Line 60: “substituted” should be “compensated”.

We changed the word to compensated.

10. Line 201: “Fishing also reduces the numbers at age in our model, leading to shorter-lived species.” What does this mean?

This is the same as previously described at comment 5., however now also including fishing mortality, which of course further reduces the numbers and thus leading that older age classes will not be reached anymore. From the example at 5, the fast growing species with low weight reached an age of 4 years (numbers ≥ 0.001), however in both the fishing scenarios only to age 3 (numbers ≥ 0.001).

Reviewer II

Färber et al. use an age-structured model of a fish population, accounting for how its dynamics vary with body size, to analyze the sensitivity of the population’s vital rates to environmental variability with different degree of autocorrelation (white/red noise) and harvesting (low/high). They do so by using a model published by Andersen & Beyer (2015) and incorporate environmental variability, and slightly reparameterizing the model. They calculate variability over 499 time steps for a range of combinations of asymptotic size and growth rates. They also chose five marine fish species (which differ to varying degree in their asymptotic size and growth rates), and subject these to two levels of fishing, combined with the two types of noise, to study the effect on recruitment. They conclude that the least variable fish stocks are those with large asymptotic sizes and at the same time fast growing life histories, and that fishing increases population variability.

The study addresses a common and well-studied aspect of population dynamics: how populations respond dynamically to environmental variation, how it propagates to later life stages than the one it directly acts on, and the interactive effects of mortality on these later stages. However, the introduction fails to recognize the previous literature in this field (its vast, some suggestions on literature are included below). Without a sufficient background on earlier studies of environmental noise of different color and how it together with mortality shapes population dynamics, it is hard to see which the current knowledge gap is, and how the current study may fill that. For example, previous studies have already addressed the interactive effect of fishing and noise colour (acting on recruitment, as in this study) (e.g., Kuparinen et al 2014). So, it is unclear where the novelty lies in the work by Färber et al, and the authors fails to convincingly demonstrate that in their introduction.

We highlight in the introduction now more the previous studies and highlight what the differences are to our study conducted here:

Our study shows many varying life-history parameters, due to fitting the model to Fishbase data and extrapolating to a large grid, covering these life-history traits. Thus, we can make more general conclusions over a large and biologically realistic parameter space, than just looking at for example cod as in Kuparinen et al. (2014)). The 5 species we select are only

examples, and they are relevant from a management point of view due to their importance to fisheries, however it could have been any other type of species.

We made a statement now in the last paragraph of the introduction: “*Here, we included the environmental noise on the recruitment efficiency in order to explore the effect of recruitment variability on spawning stock biomass across a broad range of fish life histories with the aim of identifying general patterns. In particular, we investigate how early-life density dependence, in the form of a stock-recruitment relationship, can buffer these varying environmental effects in fished populations.*” (L111 ff.)

It may be that the role of growth and body size for population dynamical responses to noise color and mortality has not been previously demonstrated. If so, I urge the authors to carefully review and synthesize available studies on noise color and population dynamics to demonstrate this. And, to relate to these and in your revised discussion clearly show how your study complements them.

There are studies indicating the importance of growth rate (e.g. (Juan-Jordá et al., 2015) and size (e.g. female size and age to buffer environmental effects (e.g. Perry et al., 2010; Rouyer et al., 2011) in the response to of fish to variability and or fishing pressure and we highlight these studies in the discussion. However, since our model is more general and not focused on a few species, we can make general conclusions on the effect of environmental noise on fish stocks (see revised discussion).

The presentation of methods and results is also somewhat unclear, making it hard to follow the set-up and part of the findings. The method section lacks a description of which analyses that were made using the model. From the results text and figure 3 (legend vs axis label) it is unclear whether it is both the level and the variation of recruitment that was studied. The authors also make conclusions on mechanisms underlying the observed patterns (e.g. that there is a buffering effect of the large body size) without supporting these by any analyses.

The buffering effect of large body sizes is visible in the decreased spread in recruitment values in response to the environmental noise and we show this in the boxplots where the range of observed recruitment values is much smaller for a species like tuna than for example for herring. We adjusted the Figure legend 3 and y-axis, which is now only labeled “Recruitment” and the figure legend reads:

“Boxplots of response of recruitment to environmental disturbance and/or fishing of 5 selected species with varying life histories within the parameter space of growth and asymptotic size. Anchovy: $W_{\infty} \approx 49$ g, $A \approx 13.4$ g^{1-n} year⁻¹; Atlantic cod: $W_{\infty} \approx 14$ kg, $A \approx 5.9$ g^{1-n} year⁻¹; Atlantic herring: $W_{\infty} \approx 412$ g, $A \approx 5.9$ g^{1-n} year⁻¹; Golden redfish: $W_{\infty} \approx 1.7$ kg, $A \approx 1.8$ g^{1-n} year⁻¹, and Yellowfin tuna: $W_{\infty} \approx 59$ kg, $A \approx 19$ g^{1-n} year⁻¹. Their respective spread in recruitment (can be seen as variability) is plotted for each of the scenarios. The three boxplots on the left of each respective species indicates scenarios with white noise with no fishing, with 0.3 year⁻¹ fishing mortality and 1.2 year⁻¹ fishing mortality; followed by the three scenarios with red noise. Note the varying y-axis scales.”

Similarly, they also suggest that their results may not be robust to the setup of their noise effect (that it acts on recruitment), but do not test this in their model (i.e. adding noise to other population processes).

The scope of this paper was limited on the recruitment dynamics and the survival of early life stages is the main driver of marine species, thus we did not include further analyses, since they would be outside the scope of the paper. However, we highlight in the discussion that this could be interesting topics for further explorations.

“However, since, the main response of (marine) fishes to environmental variability lies in the survival of the early life stages which then translates into recruitment [12], we limited, in this study, the investigations on these dynamics. However, this would be an interesting topic for further investigations in the future.” (L296 ff.)

Minor comments reviewer II

L24: “fluctuations in recruitment, which propagates throughout harvested biomass” sounds strange, as the harvested biomass is the killed biomass. So either change to “harvestable biomass” or “propagates to harvested biomass”

L30 ff. and following changed now to: “The variability in the environment often translates into fluctuations in recruitment, which then propagate throughout the stock biomass”

L 92 & tables; suggest you switch order of Table 1 & 2, as the parameters (Table 1) are only interpretable in the light of the equations (Table 2).

Tables are changed in the numbering.

L95 life table is not a method for time integration, suggest you write matrix model instead.

Changed to “*matrix model*” in the manuscript (Line 125)

L104 suggest you remove “(freshwater or marine)”; this classification is irrelevant as origin is irrelevant for your question and model set-up

Removed freshwater or marine (compare L135-136).

L110: these are usually written with a vector of abundances multiplied by a projection matrix

The matrix formulation is modified now (see L147)

L142: remove excess “the”

Removed.

L142: state the date when you downloaded data from Fishbase

Data was downloaded again during the revision process, it is now written in the manuscript “*Downloaded the available data in July 2019*” (L132)

L142: what parameter space did you obtain from Fishbase (for which parameters)? And what did you plot in there and why? Are you referring to the species-specific values of growth rates, asymptotic weights and maturation ages? If so, clarify that in the first sentence bringing this up.

We took the values for asymptotic length and growth rate from Fishbase and calculated from those the asymptotic weight and growth rate A. We then used those values to plot the ranges experienced by 5 species (within 5 fish families) into our variability plot (Figure 2 and Appendix Figure A3).

The manuscript reads now: *“For this study, we investigated a parameter space within the 80 % most common asymptotic sizes and growth rates of the species reported included in the FishBase database [...]”* (L129)

L143-144: “data was plotted as a convex hull”; unclear why (and which) data was plotted like this.

We specified now in the manuscript: *“The FishBase data from the five fish families was plotted as a convex hull (the smallest possible area that contains all the data points)”* (L191)

L147-148: “most harvested species” change to “most intensely harvested species”

Changed to most intensely harvested species (see L194).

L160 ff here I’m missing some explanations of which analyses you made in the model

We tried to be more clear which analyses were done and rewrote the method section.

L164 to support the conclusion that the recruitment variability imposed by the noisy mortality is detectable also in SSB this needs to be shown (in appendix figs) for all species

We added a figure into the appendix (SSB & Recruitment variability) (Figure A1) and write in the result section:

“This variability then translated from recruitment through the whole stock and was detectable in the SSB (Figure 1, Figure A1)” (L198 f.)

L174 rephrase to “Herring showed larger recruitment variability in response to environmental noise than the other five...”; please add figure reference for this statement

Figure reference should be to the boxplot (Figure 3, and see also A1) and is included now in the sentence which is slightly modified now to: *“Cod showed lower variability in recruitment than e.g. smaller sized species such as herring (Figure 3, Appendix Figure A1) despite having similar growth rates (Table 3)”* L208 f.

L177-L183 if Golden redfish and Atlantic cod have similar life histories, what is the explanation to the differences in how they respond to fishing? But, judging from fig 2, Golden redfish is about 10 times smaller than cod and has about 5 times slower growth rate. Isn't it? If so, could that explain their different responses.

This was a misleading formulation of the sentence. The paragraph is modified now (see L212 ff).

L186-188 From figure A1 it seems as if the panels with red noise have consistently darker color (higher CV of recruitment) than those with white noise, at least for the cases with fishing. So didn't it have an effect on population variability? Maybe use a different color scheme for your CV-levels would make it easier to see?

Now with the new analysed data there is no clear difference visible anymore, but in scenarios with red noise there is slightly more species that collapsed (Fig 3 and A2)

L226-229 this motivation should be presented already in the methods (and, preferably also in the introduction)

Included in the introduction, L111 ff

L229-231 yes, previous studies have shown that in which life stage the environmental noise is included has strong effects on the outcome. Do your conclusions hold if the noise is added to another life-stage process? Show (by supplementary analyses) or motivate.

We wanted to model the response of the recruitment to disturbances and environmental noise has strong effects on the recruitment of fish species, thus we did not model it on other life stages. (see comment to general comment above). We have it included a statement in the discussion. (L296 ff.)

L232 please expand and clarify what these could be, based on previous studies. Alternatively, include these analyses in the supplement.

We clarified now, what this could be. *“More variability, also in the fishing could have led to different outcomes, for example, due to delayed management action, when stocks are declining (due to the combination of bad environmental conditions and fishing), fishing mortality could have been intensified at low biomasses. This could have led to more collapses in stocks.”* (L 291 ff.)

L237-239 you do not have freshwater/marine water in your model, so what's the explanation in your model? Link the smaller size (and their growth rate) to what red noise does to the dynamics.

We removed this part from the discussion, since it is not relevant anymore. We added in the discussion a specific part considering the role of growth rate to the resilience of species: *“In our model, the noise colour did not lead to strong differences in the response of the stocks, however slightly more stocks collapsed/displayed larger variability under red*

noise conditions, which could be expected from the literature (e.g. Kuparinen et al., 2014; Lawton, 1988; Ruokolainen et al., 2009; Steele and Henderson, 1984). Especially, slower growing species, in the scenarios with fishing reacted stronger on red noise. This is due to the possible increased pressure of several bad years of recruitment, due to the red noise autocorrelation (Lawton, 1988), leading, next to fishing, to an additional reduction in stock size and make the stock more vulnerable. Increased vulnerability due to slower growth rate was for example also shown in tuna species/populations (Juan-Jordá et al., 2015)” (L312 ff.)

L245-247 didn't you test whether that statement holds? You did runs with higher recruitment efficiency in fig A2. Suggest you refer to that here.

We removed parts of this discussion, since it is not relevant anymore; however, we refer to our sensitivity analysis of the parameters, to show that changes in the parameters do not affect the results greatly (Appendix Figure A5)

L429 table legend hard to understand. Isn't it just model equations? If so, I suggest you rephrase to Model equations.

We modified the legend of the table. It reads now: “Parameters used in the equations, noted in Table 1.”

Figure 3: if this is depicting recruitment variability, what does the y-axis label mean? Isn't this a proportion (R/Rmax)? So isn't the variability shown as the range of the boxplot increases? In that case, the legend needs to be corrected (now reads “boxplots of recruitment variability”, which means that the boxplots then depict the variation in that variability). If the legend is correct, then the y-axis labels need to be corrected.

See response to general comment above and newly formulated legend of Figure 3:

“Boxplots of response of recruitment to environmental disturbance and/or fishing of 5 selected species with varying life histories within the parameter space of growth and asymptotic size. Anchovy: $W_{\infty} \approx 50$ g, $A \approx 14$ g l⁻ⁿ year⁻¹; Atlantic cod: $W_{\infty} \approx 12$ kg, $A \approx 5.6$ g l⁻ⁿ year⁻¹; Atlantic herring: $W_{\infty} \approx 400$ g, $A \approx 5.2$ g l⁻ⁿ year⁻¹; Golden redfish: $W_{\infty} \approx 1.7$ kg, $A \approx 1.7$ g l⁻ⁿ year⁻¹, and yellowfin tuna: $W_{\infty} \approx 48$ kg, $A \approx 22$ g l⁻ⁿ year⁻¹. Their respective spread in recruitment (can be seen as variability) is plotted for each of the scenarios. The three boxplots on the left of each respective species indicates scenarios with white noise with no fishing, with 0.3 year⁻¹ fishing mortality and 1.2 year⁻¹ fishing mortality; followed by the three scenarios with red noise. Note the varying y-axis scales.”

References

- Andersen, K. H. 2019. *Fish Ecology, Evolution, and Exploitation-A New Theoretical Synthesis*, Princeton University Press.
- Andersen, K. H., and Beyer, J. E. 2015. Size structure, not metabolic scaling rules, determines fisheries reference points. *Fish and Fisheries*, 16: 1-22.
- Beverton, R. J. H. 1992. Patterns of reproductive strategy parameters in some marine teleost fishes. *Journal of Fish Biology*, 41: 137-160.
- Dickey-Collas, M., Nash, R. D. M., Brunel, T., van Damme, C. J. G., Marshall, C. T., Payne, M. R., Corten, A., et al. 2010. Lessons learned from stock collapse and recovery of North Sea herring: a review. *ICES Journal of Marine Science*, 67: 1875-1886.
- Froese, R. 2006. Cube law, condition factor and weight-length relationships: history, meta-analysis and recommendations. *Journal of Applied Ichthyology*, 22: 241-253.
- Froese, R., and Pauly, D. 2018. *FishBase*. World Wide Web electronic publication. www.fishbase.org, version(06/2018).
- Gislason, H., Daan, N., Rice, J. C., and Pope, J. G. 2010. Size, growth, temperature and the natural mortality of marine fish. *Fish and Fisheries*, 11: 149-158.
- Gunderson, D. R. 1997. Trade-off between reproductive effort and adult survival in oviparous and viviparous fishes. *Canadian Journal of Fisheries and Aquatic Sciences*, 54: 990-998.
- Hartvig, M., Andersen, K. H., and Beyer, J. E. 2011. Food web framework for size-structured populations. *Journal of Theoretical Biology*, 272: 113-122.
- Hyllen, A., Nakken, O., and Nedreaas, K. 2008. Northeast Arctic cod: fishery, life history, stock fluctuations and management. *In Norwegian Spring-spawning Herring & Northeast Arctic Cod: 100 Years of Research Management*, pp. 83-118. Ed. by O. Nakken. Tapir Academic Press, Trondheim.
- Juan-Jordá, M. J., Mosqueira, I., Freire, J., and Dulvy, N. K. 2015. Population declines of tuna and relatives depend on their speed of life. *Proceedings of the Royal Society B: Biological Sciences*, 282: 20150322.
- Kokkalis, A., Eikeset, A. M., Thygesen, U. H., Steingrund, P., and Andersen, K. H. 2016. Estimating uncertainty of data limited stock assessments. *ICES Journal of Marine Science*, 74: 69-77.
- Kuparinen, A., Keith, D. M., and Hutchings, J. A. 2014. Increased environmentally driven recruitment variability decreases resilience to fishing and increases uncertainty of recovery. *ICES Journal of Marine Science*, 71: 1507-1514.
- Lawton, J. H. 1988. More time means more variation. *Nature*, 334: 563.
- Levins, R. 1966. The strategy of model building in population biology. *American Scientist*, 54: 421-431.
- Ottersen, G., Kim, S., Huse, G., Polovina, J. J., and Stenseth, N. C. 2010. Major pathways by which climate may force marine fish populations. *Journal of Marine Systems*, 79: 343-360.
- Perry, R. I., Cury, P., Brander, K., Jennings, S., Möllmann, C., and Planque, B. 2010. Sensitivity of marine systems to climate and fishing: Concepts, issues and management responses. *Journal of Marine Systems*, 79: 427-435.
- Rouyer, T., Ottersen, G., Durant, J. M., Hidalgo, M., Hjermann, D. Ø., Persson, J., Stige, L. C., et al. 2011. Shifting dynamic forces in fish stock fluctuations triggered by age truncation? *Global Change Biology*, 17: 3046-3057.
- Ruokolainen, L., Lindén, A., Kaitala, V., and Fowler, M. S. 2009. Ecological and evolutionary dynamics under coloured environmental variation. *Trends in Ecology & Evolution*, 24: 555-563.

Steele, J. H., and Henderson, E. W. 1984. Modeling Long-Term Fluctuations in Fish Stocks. *Science*, 224: 985-987.

West, G. B., Brown, J. H., and Enquist, B. J. 1997. A General Model for the Origin of Allometric Scaling Laws in Biology. *Science*, 276: 122-126.

Appendix C

We would again like to thank the editors and reviewers for constructive and helpful comments that have helped us to further improve the manuscript. Find our detailed reply to the individual comments directly below each comment in bold font.

Associate Editor Comments to Author (Dr Punidan Jayasingh):

I thank the authors for a thorough revision. This version is much improved, and was assessed by two experts. While the reviews were more favorable this time around, both reviewers felt that the language and presentation could be further improved. The reviewers make a number of recommendations to mitigate the issue. I felt the reviews were fair and constructive. I recommend the authors make these revisions and send the manuscript to an anglophone (ideally one who isn't an expert in the field) for comments to improve comprehensibility before submitting another version. The science is strong, however, the work needs to be presented in a clearer fashion. I look forward to reading a revised version.

Thanks for your comments and agreeing to read once more our work. We have addressed all the comments of the reviewers, see below for details. Furthermore, to improve the English language, we have used a certified language editing service (Charlesworth), see attached document for more information.

Reviewer comments to Author:

Reviewer: 3
Comments to the Author(s)

In general, I found the author responses thorough and well motivated. Therefore, I recommend a minor revision addressing the minor errors I found in the paper.

Thank you for the positive evaluation of our manuscript.

The only major thing that would have been interesting to add to the current manuscript is an analysis showing the effect of noise on other processes in the model (even though recruitment may be the most uncertain process in the model, other parts of the recruitment may equally likely be affected (like through R_{max} or a direct additive effect on R_t)). This would probably have major implications for responses with respect to the different noise processes (white and red noise). However, I think the authors have already done a thorough job in addressing all reviewer comments and therefore accept if no such analysis is conducted.

We did not perform such an analysis as we consider this outside the scope of the current manuscript. We agree with the reviewer that noise can also affect other processes, such as R_{max} or R_t . However, the main effect of noise is likely to affect the recruitment process, and we prefer to focus on this part of the model. This is clarified in the text by:

“We included the impact of environmental noise on the recruitment efficiency in order to investigate the effect of recruitment variability on the spawning stock biomass, and examine how early-life density-dependence, in the form of a Beverton-Holt stock-recruitment relationship, buffers this effect. The impact of environmental noise on maximum recruitment (representing the early-life carrying capacity in a certain environment) [59], or on the survival of the adult stock, could also have been included, but was not the focus in this study.”

Here are some minor comments:

* Equation numbers should be corrected.

The equations are numbered in order of appearance in the text.

* There is a time index missing in the second part of eq.(2).

We have now added the correct index to equation (2) both in the text and Table 1.

* Inequalities with wrong direction (l. 204 & l. 205)

We have corrected these mistakes.

* (Eq.(12) in table 1) wrongly specified model of red noise. It should read: $x_{red}(t) = c * x_{red}(t-1) + x_{white}(t) * \sqrt{1 - c^2}$

Thanks for pointing out this mistake. We have corrected the equation in table 1.

* Please state in the figure legend to figure 2 the response variable being used. Is it SSB or recruitment?

We have clarified that the response variable is the recruitment.

Reviewer: 4

Comments to the Author(s)

I found the paper difficult to follow. One reason for this is that a lot of the sentences are long and unclear. This may often be language issue that can relatively easily be solved, but may also be due to unclear thinking.

Thank you for the evaluation of the manuscript. We have made a major effort to improve the language of the manuscript. This includes answering the detailed comments below, a thorough language editing by the authors as well by a professional language editing service. We believe that these efforts have clarified and alleviated the difficulties of following the paper. See below for further details.

Details

- In the title, abstract and introduction you use the terms growth (and growth rate) and asymptotic size. It wasn't until well into the methods that I became confident that it is body mass growth and asymptotic body mass (size) that was meant. Make this clear already in the title, abstract and introduction. (The way the expressions are used, you could have been referring to population growth and a parameter for some asymptotic population size).

We have clarified in the title, abstract and introduction that we mean body mass growth and asymptotic body mass.

- Abstract line 32-35. Long awkward sentence. Rewrite.

We have split and rewritten the sentence. It now reads:

"Here, we systematically explore the dynamics and sensitivity of fish stock recruitment and biomass to environmental noise. Using an age-structured and trait-based model, we explore random noise (white noise) and autocorrelated noise (red noise) in combination with low to high levels of harvesting."

- Line 37-38. I find it strange to claim that "Our study indicates that the variability of stock recruitment and biomass can be driven by the stock's asymptotic size and growth rate". The body mass growth parameters you focus on do not drive variability, but may be associated with patterns in the population variability driven by environmental variability.

- Line 38-40. Fine, but cut line 37-40 into at least two sentences.

We have split and rewritten the sentence mentioned in the two comments above to:

"Our study indicates that the variability of stock recruitment and biomass are likely correlated with the stock's asymptotic size and growth rate. We find that fast-growing and large-sized fish stocks are likely to be less vulnerable to disturbances than slow-growing and small-sized fish stocks."

- Line 40-41. Is this a general result? If so make it clear that it is independent of body mass growth parameter values.

No longer relevant, the sentence has been removed.

- Line 41-43. Delete? Seems to be repetition + I am not convinced by the claim that your focal body growth parameters are "crucial for a species resilience toward s disturbances". Rework: "expected to be important predictors of" rather than "crucial for" maybe?

Done.

- Make clear what is the new knowledge gained from the work.

We followed the specific suggestion below and are explicit about the novelty in the introduction. We have also added the following sentence to the abstract.

"We show how the natural variability in fish stocks is amplified by fishing, not just on one stock, but for a broad range of fish life histories."

Introduction

- Line 57-91. I find most of the first part of the introduction difficult to read. I can guess what the authors

want to say, but the text is almost incomprehensible. Cut the text into more and simpler sentences and check that the words used are appropriate. The information should lead up to some conclusion. This part need a proper job.

E.g. I believe the second sentence can be deleted (line 57-60). The third sentence could be cut down to "The early life stages of fish are particularly sensitive to environmental variation, and this sensitivity can result in large variability in the size of the spawning stock and the stock available for the fisheries."

We have tried to simplify and clarify our text.

- It is not clear what new knowledge we gain from the study. The last few sentence do not make this clear even though the authors claim they do so in the response to reviewer II. Be direct. E.g. Previous studies have shown....Our study expand on this by showing that....

We have added the following sentence at the end of the introduction:

"Additionally, we systematically explore the roles played by growth and body size in population dynamical responses to noise colour and mortality for a broad range of fish life histories, and demonstrate that the natural variability in fish stocks is amplified by fishing."

Methods

- Line 119-120. Delete sentence.

Done.

- Line 120-129. The text need some work.

We have rewritten this part.

- Table 2. Include a definition of the index i .

We have done it for both Table 1 and 2.

- Line 140. The age index is wrong in for the abundance the next year $N_{i,t+1}$ should be $N_{i+1,t+1}$

We have corrected the sentence. It reads: "Each year $t + 1$ surviving individual of weight w_t s enters the next age class, and reproduces when mature."

- Line 151-152. Poor sentence.

We have rewritten the sentence: "Where $N_{i,t}$ is the abundance at age i and year t , and the weight-specific survival S_i (Eq. (3), Table 1) is calculated from natural mortality (Eq. (4), Table 1) and fishing mortality."

- Delete eq 7 and 8 in the main text. Refere to the equations in the Table 1.

We have deleted the equations as suggested.

- Use the same expressions for the parameters throughout. Eg. In table 2 A is called "Growth costant" while in most of the manuscript you call it growth or growth rate. I suggest "body mass growth rate"

We have replaced in the text to "growth rate" and "body mass growth rate" when appropriate.

- Line 181-183. Unclear sentence. Also, what did you do when populations crashed.

We have rewritten this sentence:

"We excluded populations with unrealistic low recruitment (with a mean recruitment remaining under 0.01) as they did not yield reliable results."

- Table 3. Include common names in addition to latin species and family names. I did not find coloumn 3 and 5 particularely interesting.

Done. We kept the two columns.

Results

- Line 204-205. Seems like > in brackets should be <.

Done.

- Line 213-216. Long sentence. Rewrite.

We have rewritten the sentence to:

“For slow-growing species, such as the golden redbfish (Figures 2 and 3), increasing fishing pressure caused a significant strong increase in recruitment variability ($p < 0.001$), with higher fishing pressure leading to a collapse.”

- Line 218. Reword “disturbances” “variability”?

Done.

- Line 221-224. Long sentence.

We have split the sentence, which now reads:

“There were no clear differences between the variability induced by white or red noise (Figure 3, Appendix Figure A3).”

Discussion

- Line 230-234. Long winded sentence. Also the 20% variability must depend on the variance in the environment used in the simulations.

The sentence now read:

“However, species with an asymptotic weight of around 140 g and a growth rate of $> 3 \text{ g} \cdot 0.25 \text{ year}^{-1}$ showed less than 20% variability ($\log_{10} = -0.7$) in their recruitment in the present simulation. This indicates that large-sized species are better able to buffer environmental effects than smaller-sized and slower growing species (Figure 2, Appendix A3).”

- Line 245. Reword “fewer”

Reworded to “less”.

- Line 247-249. You haven't shown a reduction in the number of cohorts in the populations associated with fishing.

We have rewritten the sentence to:

“However, we find, as predicted by several other studies, that the erosion of the stock structure (e.g. by fishing) may lead to reduced capacity of the stock to buffer disturbances.”

- Line 250-251. Unclear sentence.

Reworded sentence now reads:

“However, fast-growing and large-sized species are more capable of buffering against these effects (Figure A4).”

- Line 263-265. Unclear sentence.

We have changed the sentence to:

“In many stocks, the maternal effects associated with older fish do not significantly affect recruitment success [47, 48] and hence do not affect the overall abundance of the stock.”

- Line 266-269. Long unclear sentence. It is unclear what is eroded.
- Line 269-270. Poor sentence. The term overfishing imply sensitivity to fishing.

The sentences now read:

“Species with high population biomass due to large asymptotic weight and rapid growth, such as the tuna, likely have to be subject to substantial erosion of age or size structure (i.e. exceeding the value applied in our scenarios of fishing mortality) to show an effect on recruitment variability; however, yellowfin tuna has apparently been subjected to overfishing.”

- Line 270-271. Poor sentence. (change “the decreased biomass” to “a decreased biomass”?)

Done.

- Line 273. Not “inhabited”, wrong word.

Changed to “inhibited”.

- Line 274-277. Poor sentence. What is an “older population structure” ?

We have split the sentence in two and clarified what we meant by an “older population structure”.
“Similarly, many small-sized pelagic fishes, such as herring, experience high fishing pressure and, in general, exhibit stronger fluctuation in recruitment with environmental conditions. However, they seem to recover relatively faster by regaining or maintaining a population structure that includes old individuals.”

- Line 282. Reword “sustain”

Done, the sentence now reads:

“The importance of growth rate as a buffer against environmental variability can also be seen in various species of the Scombridae family.”

- Line 287. Reword “on” to “about”?

Done.

- Line 292-296. Poor sentence.

We have rewritten the sentence to:

“Increased variability, e.g. in fishing pressure, could have altered the outcomes. For example, in declining stocks, fishing mortality may potentially be intensified at low biomasses due to delayed management action. This may lead to increased probability of collapse in stocks.”

- Line 306-309. Poor sentence.

We have changed the sentence to:

“Increased variability, e.g. in fishing pressure, may have altered the outcomes. For example, in declining stocks, fishing mortality may potentially be intensified at low biomasses as a result of delayed management action. This may lead to increased probability of collapse in stocks.”

- Line 311-313. Long poor sentence. Delete?

We agree and have deleted this sentence.

- Line 317-319. Long poor sentence.

The sentence now reads:

“This may potentially be attributable to the increased pressure of several bad years of recruitment caused by the red noise autocorrelation. A sequence of weak year-classes may, combined with fishing, lead to a severe reduction in stock size and make the stock more vulnerable to collapse.”

- Line 327-329. Poor sentence.

We have rewritten the sentence to:

“Finally, as previously reported, we find that a healthy age structure may buffer environmental effects and increase the resilience to disturbances at the stock level.”

- Legend to figure 2. Needs tightening up. The parameters for the species are already given in table 3 and do not need to be repeated (refer to table 3).

The figure caption has been tightened up, including removal of the parameter values and some simplification of the description of the figure.

- Legend to figure 3. The parameters for the species are already given in table 3 and do not need to be repeated (refer to table 3).

We have removed the information and now refer to Table 3.